# WYNER VAE: A VARIATIONAL AUTOENCODER WITH SUCCINCT COMMON REPRESENTATION LEARNING

## ABSTRACT

A new variational autoencoder (VAE) model is proposed that learns a succinct common representation of two correlated data variables for conditional and joint generation tasks. The proposed Wyner VAE model is based on two information theoretic problems—distributed simulation and channel synthesis—in which Wyner's common information arises as the fundamental limit of the succinctness of the common representation. The Wyner VAE decomposes a pair of correlated data variables into their common representation (e.g., a shared concept) and local representations that capture the remaining randomness (e.g., texture and style) in respective data variables by imposing the mutual information between the data variables and the common representation as a regularization term. The utility of the proposed approach is demonstrated through experiments for joint and conditional generation with and without style control using synthetic data and real images. Experimental results show that learning a succinct common representation achieves better generative performance and that the proposed model outperforms existing VAE variants and the variational information bottleneck method.

## 1 INTRODUCTION

This paper aims to develop a new probabilistic framework for generation tasks (i.e., learning the distribution of given data and sampling from the learned distributions) for two high-dimensional random vectors. To motivate the main idea, consider the following cooperative game between Alice and Bob. Suppose that given an image of a child's photo, Alice sends its description $\mathbf{Z}$ to Bob who draws a portrait of how the child will grow up based on it. The objective of this game is to draw a nice portrait, and thus Alice needs to help Bob in the process by providing a *good* description of the child's photo — any redundant information in the description may confuse Bob in his guessing process. What description does Alice need to generate and send from the child's photo?

P. Cuff (2013) formulated this game of conditional generation as the *channel synthesis* problem in network information theory depicted in Fig. 1. Given a joint distribution $q(\mathbf{x}, \mathbf{y}) = q(\mathbf{x})q(\mathbf{y}|\mathbf{x})$, Alice and Bob want to generate $\mathbf{Y}$ according to $q(\mathbf{y}|\mathbf{x})$ based on a sample from $q(\mathbf{x})$. In this problem, Alice wishes to find the most succinct description $\mathbf{Z}$ of $\mathbf{X}$ (a child's photo) such that $\mathbf{Y}$ (her adulthood portrait) can be simulated by Bob according to the desired distribution using this description and local randomness $\mathbf{V}$ (new features to draw a portrait of adults that are not contained in photos of children). The minimum description rate for such conditional generation is characterized by Wyner's *common information* (Wyner, 1975; El Gamal and Kim, 2011) denoted by $J(\mathbf{X}; \mathbf{Y})$ and defined as the optimal value of the optimization problem

$$
\begin{aligned}
&\text{minimize} && I_q(\mathbf{X}, \mathbf{Y}; \mathbf{Z}) \\
&\text{subject to} && \mathbf{X} \to \mathbf{Z} \to \mathbf{Y} \\
&\text{variables} && q(\mathbf{z}|\mathbf{x}, \mathbf{y}),
\end{aligned}
\tag{1}
$$

where $\mathbf{X} \to \mathbf{Z} \to \mathbf{Y}$ denotes a Markov chain from $\mathbf{X}$ to $\mathbf{Z}$ to $\mathbf{Y}$ and $I_q(\mathbf{X}, \mathbf{Y}; \mathbf{Z})$ denotes the mutual information between $(\mathbf{X}, \mathbf{Y})$ and $\mathbf{Z}$, where $(\mathbf{X}, \mathbf{Y}, \mathbf{Z}) \sim q(\mathbf{x}, \mathbf{y})q(\mathbf{z}|\mathbf{x}, \mathbf{y})$.

The same quantity $J(\mathbf{X}; \mathbf{Y})$ arises as the fundamental limit of the *distributed simulation* of correlated sources studied originally by A. Wyner (1975) in which two distributed agents wish to simulate a target distribution $q(\mathbf{x}, \mathbf{y})$ (i.e., joint generation of $(\mathbf{X}, \mathbf{Y})$) based on the least possible amount of

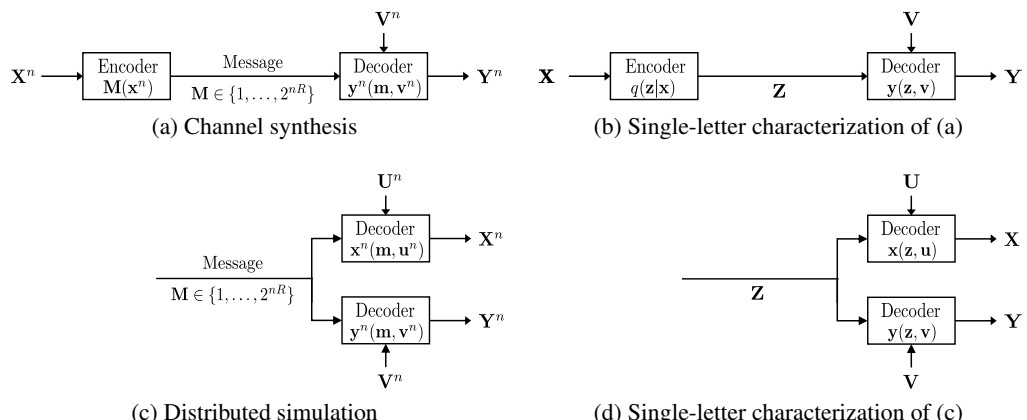

Figure 1: Schematics for channel synthesis from $\mathbf{X}$ to $\mathbf{Y}$ (a,b), and distributed simulation of $(\mathbf{X}, \mathbf{Y})$ (c,d). (a,c) and (b,d) correspond to the operational definition and the single-letter characterization of each problem, respectively. The local randomness $\mathbf{U}$ and $\mathbf{V}$ make the decoders stochastic.

shared common randomness. (See Fig. 1 (c,d).) In this sense, the joint distribution $q(\mathbf{x}, \mathbf{y})$ and the conditional distributions $q(\mathbf{y}|\mathbf{x})$, $q(\mathbf{x}|\mathbf{y})$ have the same common information structure characterized by the optimization problem (1), which involves learning the joint distribution in its nature. We call the joint encoder $q(\mathbf{z}|\mathbf{x}, \mathbf{y})$ (or equivalently, the corresponding random variable $\mathbf{Z}$) as the *common representation* of $(\mathbf{X}, \mathbf{Y})$, and the mutual information $I_q(\mathbf{X}, \mathbf{Y}; \mathbf{Z})$ then can be viewed as a measure of the complexity of $\mathbf{Z}$.

The goal of this paper is to propose a new probabilistic model based on these information theoretic observations, to achieve a good performance in joint and conditional generation tasks. We apply the idea of learning succinct common representation to design a new generative model for a pair of correlated variables: seeking a succinct representation $\mathbf{Z}$ in learning the underlying distribution based on its sample may also help reduce the burden on the decoder's side and thereby achieve a better generative performance.

The rest of the paper gradually develops our framework as follows. We first define a probabilistic model based on the motivating problems, which we aim to train and use for generation tasks (Section 2.1), and then establish a general principle for learning the model based on the optimization problem (1) (Section 2.2). We propose one instantiation of the principle with a standard variational technique by introducing additional encoder distributions (Section 2.3). The proposed model with its training method can be viewed as a variant of variational autoencoders (VAEs) (Kingma and Welling, 2014; Rezende et al., 2014), and is thus called *Wyner VAE*. (See Appendix A for a brief introduction on VAEs.) The new encoder components introduced in Wyner VAE allow us to decompose a data vector into the common representation and the local representation, which can be used for sampling with style manipulation (Section 2.4). We carefully compare our model and show its advantages over the existing VAE variants (Vedantam et al., 2018; Suzuki et al., 2016; Sohn et al., 2015; Wang et al., 2016) and the information bottleneck (IB) principle (Tishby et al., 1999) (Section 3), which is a well-known information theoretic principle in representation learning. In the experiments, we empirically show the utility of our model in various sampling tasks and its superiority over existing models and that learning a succinct common representation achieves better generative performance in generation tasks (Section 4).

## 2 Wyner variational autoencoder

### 2.1 Probabilistic model

We first define a probabilistic model for joint and conditional sampling tasks based on the single-letter characterizations of the motivating problems (Fig. 1 (b,d)). We assume that all distributions to be

introduced below belong to some standard parametric families such as Gaussians, and use $(q, \phi)$ (or $(p, \theta)$) to denote the parameters and the distribution of encoders (or decoders and priors).

In both channel synthesis and distributed simulation, $\mathbf{Z} \sim p_\theta(\mathbf{z})$ signifies the common randomness fed into the deterministic decoders $\mathbf{x}_\theta(\mathbf{z}, \mathbf{u})$ and $\mathbf{y}_\theta(\mathbf{z}, \mathbf{v})$, while $\mathbf{U} \sim p_\theta(\mathbf{u})$ and $\mathbf{V} \sim p_\theta(\mathbf{v})$ signify the local randomnesses for each decoder. We take the sources of randomness (or priors) $p_\theta(\mathbf{z}), p_\theta(\mathbf{u}), p_\theta(\mathbf{v})$ as standard parametric distributions such as Gaussian from which a sample can be easily drawn. $\mathbf{U}$ (or $\mathbf{V}$) is interpreted as a local description of $\mathbf{X}$ (or $\mathbf{Y}$) given the common description $\mathbf{Z}$. We can also view $\mathbf{U}$ (or $\mathbf{V}$) as an intrinsic randomness of the stochastic decoder $\tilde{p}_\theta(\mathbf{x}|\mathbf{z})$ (or $\tilde{p}_\theta(\mathbf{y}|\mathbf{z})$), which is the *induced* distribution by the decoder $\mathbf{x}_\theta(\mathbf{z}, \mathbf{u})$ (or $\mathbf{y}_\theta(\mathbf{z}, \mathbf{v})$) and the prior $p_\theta(\mathbf{u})$ (or $p_\theta(\mathbf{v})$). To perform joint sampling, we need the priors $p_\theta(\mathbf{z}), p_\theta(\mathbf{u}), p_\theta(\mathbf{v})$ and the decoders $\mathbf{x}_\theta(\mathbf{z}, \mathbf{u})$ and $\mathbf{y}_\theta(\mathbf{z}, \mathbf{v})$ (Fig. 1 (d)). For conditional sampling of $\mathbf{Y}$ given $\mathbf{X} = \mathbf{x}$, we need the *marginal* encoder $q_\phi(\mathbf{z}|\mathbf{x})$ to be defined in addition to the prior $p_\theta(\mathbf{v})$ and the decoder $\mathbf{y}_\theta(\mathbf{z}, \mathbf{v})$ (Fig. 1 (b)). We focus on how to learn $q_\phi(\mathbf{z}|\mathbf{x})$ in what follows, since $q_\phi(\mathbf{z}|\mathbf{y})$ can be dealt with in the same way by the symmetry of the Markov chain $\mathbf{X} \to \mathbf{Z} \to \mathbf{Y}$.

After Wyner, we name the entirety of all the components (the marginal encoders $q_\phi(\mathbf{z}|\mathbf{x}), q_\phi(\mathbf{z}|\mathbf{y})$, the priors $p_\theta(\mathbf{z}), p_\theta(\mathbf{u}), p_\theta(\mathbf{v})$, and the decoders $p_\theta(\mathbf{x}|\mathbf{z}, \mathbf{u}), p_\theta(\mathbf{y}|\mathbf{z}, \mathbf{v})$) that is essential for joint and conditional sampling tasks as the *Wyner common representation model* or the *Wyner model* in short.

## 2.2 OBJECTIVE FUNCTION

We now propose an objective function for learning each component in the Wyner model based on the optimization problem (1). In practice, the data distribution $q(\mathbf{x}, \mathbf{y})$ is replaced by the empirical distribution $q_{\text{emp}}(\mathbf{x}, \mathbf{y})$ defined by the given samples. To train the components for joint sampling in the Wyner model, we solve the optimization problem (1) by incorporating a new consistency constraint

$$q(\mathbf{x}, \mathbf{y})q_\phi(\mathbf{z}|\mathbf{x}, \mathbf{y}) \equiv p_\theta(\mathbf{z})\tilde{p}_\theta(\mathbf{x}|\mathbf{z})\tilde{p}_\theta(\mathbf{y}|\mathbf{z}), \tag{2}$$

with the priors $p_\theta(\mathbf{z}), p_\theta(\mathbf{u}), p_\theta(\mathbf{v})$ and the decoders $\mathbf{x}_\theta(\mathbf{z}, \mathbf{u}), \mathbf{y}_\theta(\mathbf{z}, \mathbf{v})$ as new optimization variables. The Markov condition $\mathbf{X} \to \mathbf{Z} \to \mathbf{Y}$ can be removed under the new constraint (2). We then relax the constraint (2) with an inequality constraint

$$\mathcal{L}_{\mathbf{xy} \to \mathbf{xy}} := D(p_\theta(\mathbf{z})\tilde{p}_\theta(\mathbf{x}|\mathbf{z})\tilde{p}_\theta(\mathbf{y}|\mathbf{z}), q(\mathbf{x}, \mathbf{y})q_\phi(\mathbf{z}|\mathbf{x}, \mathbf{y})) \leq \epsilon,$$

for some $\epsilon > 0$, and convert the problem (1) into an unconstrained form as in Zhao et al. (2018):

$$\begin{aligned} \text{minimize} \quad & \mathcal{L}_{\mathbf{xy} \to \mathbf{xy}} + \lambda I_q \\ \text{variables} \quad & q_\phi(\mathbf{z}|\mathbf{x}, \mathbf{y}), p_\theta(\mathbf{z}), p_\theta(\mathbf{u}), p_\theta(\mathbf{v}), \mathbf{x}_\theta(\mathbf{z}, \mathbf{u}), \mathbf{y}_\theta(\mathbf{z}, \mathbf{v}) \end{aligned} \tag{3}$$

Here we use a shorthand notation $I_q := I_q(\mathbf{X}, \mathbf{Y}; \mathbf{Z})$, and use $\lambda > 0$ to denote the reciprocal of the Lagrange multiplier. We can choose $D(p, q)$ as any proper distance or divergence measure between distributions such as $f$-divergences, Jensen–Shannon divergence, Wasserstein distance, or maximum mean discrepancy (Zhao et al., 2018).

To find the marginal encoder $q_\phi(\mathbf{z}|\mathbf{x})$ that is consistent to $q(\mathbf{y}|\mathbf{x})q_\phi(\mathbf{z}|\mathbf{x}, \mathbf{y})$ for conditional generation, we aim to match a joint distribution induced by the Markov chain $\mathbf{X} \to \mathbf{Z} \to \mathbf{Y}$, i.e., $q(\mathbf{x})q_\phi(\mathbf{z}|\mathbf{x})\tilde{p}_\theta(\mathbf{y}|\mathbf{z})$, with the decoder distribution $p_\theta(\mathbf{z})\tilde{p}_\theta(\mathbf{x}|\mathbf{z})\tilde{p}_\theta(\mathbf{y}|\mathbf{z})$ and the encoder distributions $q(\mathbf{x}, \mathbf{y})q_\phi(\mathbf{z}|\mathbf{x}, \mathbf{y})$ of the joint model. That is, we wish to find $q_\phi(\mathbf{z}|\mathbf{x})$ that minimizes

$$\mathcal{L}_{\mathbf{x} \to \mathbf{x}} := D(p_\theta(\mathbf{z})\tilde{p}_\theta(\mathbf{x}|\mathbf{z}), q(\mathbf{x})q_\phi(\mathbf{z}|\mathbf{x})) \tag{4}$$

and

$$\mathcal{L}_{\mathbf{xy} \to \mathbf{y}} := D(q(\mathbf{x})q_\phi(\mathbf{z}|\mathbf{x})\tilde{p}_\theta(\mathbf{y}|\mathbf{z}), q(\mathbf{x}, \mathbf{y})q_\phi(\mathbf{z}|\mathbf{x}, \mathbf{y})). \tag{5}$$

The objective functions $\mathcal{L}_{\mathbf{y} \to \mathbf{y}}$ and $\mathcal{L}_{\mathbf{xy} \to \mathbf{x}}$ for $q_\phi(\mathbf{z}|\mathbf{y})$ can be defined in a symmetric manner. The final objective function for training the Wyner model with succinct common representation learning then becomes

$$\mathcal{L} := \mathcal{L}_{\mathbf{xy} \to \mathbf{xy}} + \lambda I_q + \alpha_{\mathbf{x} \to \mathbf{x}}\mathcal{L}_{\mathbf{x} \to \mathbf{x}} + \alpha_{\mathbf{xy} \to \mathbf{y}}\mathcal{L}_{\mathbf{xy} \to \mathbf{y}} + \alpha_{\mathbf{y} \to \mathbf{y}}\mathcal{L}_{\mathbf{y} \to \mathbf{y}} + \alpha_{\mathbf{xy} \to \mathbf{x}}\mathcal{L}_{\mathbf{xy} \to \mathbf{x}}, \tag{6}$$

where the weights $\alpha$'s are nonnegative hyperparameters.

Yet, to minimize the objective function in practice, we need to choose which divergence/distance metric $D(p, q)$ to use, and also need to address computationally intractable terms—the induced distributions $\tilde{p}_\theta(\mathbf{x}|\mathbf{z}), \tilde{p}_\theta(\mathbf{y}|\mathbf{z})$ and the mutual information term $I_q$—in (3), (4), and (5).

## 2.3 VARIATIONAL RELAXATION

We propose an instantiation of the objective function with a specific choice of a divergence function with a standard variational technique. We choose the metric $D(p, q)$ as the reverse KL divergence $D_{\mathsf{KL}}(q\|p)$, which is a common choice in the variational inference literature (see, e.g., Blei et al. (2017)). To remove the intractable induced distributions, we relax the objective function (3) by introducing variational encoders $q_\phi(\mathbf{u}|\mathbf{z}, \mathbf{x})$ and $q_\phi(\mathbf{v}|\mathbf{z}, \mathbf{y})$:

$$\mathcal{L}_{\mathbf{xy}\to\mathbf{xy}} = D_{\mathsf{KL}}(q(\mathbf{x}, \mathbf{y})q_\phi(\mathbf{z}|\mathbf{x}, \mathbf{y})\|p_\theta(\mathbf{z})\tilde{p}_\theta(\mathbf{x}|\mathbf{z})\tilde{p}_\theta(\mathbf{y}|\mathbf{z})) \tag{7}$$

$$\leq D_{\mathsf{KL}}(q(\mathbf{x}, \mathbf{y})q_\phi(\mathbf{z}|\mathbf{x}, \mathbf{y})q_\phi(\mathbf{u}|\mathbf{z}, \mathbf{x})q_\phi(\mathbf{v}|\mathbf{z}, \mathbf{y})\|p_\theta(\mathbf{z})\tilde{p}_\theta(\mathbf{x}|\mathbf{z})\tilde{p}_\theta(\mathbf{y}|\mathbf{z})\tilde{p}_\theta(\mathbf{u}, \mathbf{v}|\mathbf{x}, \mathbf{y}, \mathbf{z})) \tag{8}$$

$$= D_{\mathsf{KL}}(q(\mathbf{x}, \mathbf{y})q_\phi(\mathbf{z}|\mathbf{x}, \mathbf{y})q_\phi(\mathbf{u}|\mathbf{z}, \mathbf{x})q_\phi(\mathbf{v}|\mathbf{z}, \mathbf{y})\|p_\theta(\mathbf{z})p_\theta(\mathbf{u})p_\theta(\mathbf{v})p_\theta(\mathbf{x}|\mathbf{z}, \mathbf{u})p_\theta(\mathbf{y}|\mathbf{z}, \mathbf{v})) \tag{9}$$

$$=: \overline{\mathcal{L}}_{\mathbf{xy}\to\mathbf{xy}}, \tag{10}$$

where (8) follows from the chain rule and nonnegativity of KL divergence (see, e.g., Cover and Thomas (2006)), $\tilde{p}_\theta(\mathbf{u}, \mathbf{v}|\mathbf{x}, \mathbf{y}, \mathbf{z})$ denotes the induced conditional distribution by $p_\theta(\mathbf{u}), p_\theta(\mathbf{v}), p_\theta(\mathbf{x}|\mathbf{z}, \mathbf{u}), p_\theta(\mathbf{y}|\mathbf{z}, \mathbf{v})$, and $p_\theta(\mathbf{x}|\mathbf{z}, \mathbf{u})$ (or $p_\theta(\mathbf{y}|\mathbf{z}, \mathbf{v})$) denotes the distribution induced by the decoder $\mathbf{x}_\theta(\mathbf{z}, \mathbf{u})$ (or $\mathbf{y}_\theta(\mathbf{z}, \mathbf{v})$). Note that the intractable distributions $\tilde{p}_\theta(\mathbf{x}|\mathbf{z}), \tilde{p}_\theta(\mathbf{y}|\mathbf{z})$ no longer appear in the upper bound (9).

For the mutual information term $I_q = I_q(\mathbf{X}, \mathbf{Y}; \mathbf{Z})$ with $(\mathbf{X}, \mathbf{Y}, \mathbf{Z}) \sim q(\mathbf{x}, \mathbf{y})q_\phi(\mathbf{z}|\mathbf{x}, \mathbf{y})$, we use the following standard upper bound (see, e.g., Zhao et al. (2018)):

$$I_q = \mathbb{E}_{q(\mathbf{x},\mathbf{y})}[D_{\mathsf{KL}}(q_\phi(\mathbf{z}|\mathbf{X}, \mathbf{Y})\|\tilde{q}_\phi(\mathbf{z}))] \leq \mathbb{E}_{q(\mathbf{x},\mathbf{y})}[D_{\mathsf{KL}}(q_\phi(\mathbf{z}|\mathbf{X}, \mathbf{Y})\|p_\theta(\mathbf{z}))] =: \overline{I}_q, \tag{11}$$

where $\tilde{q}_\phi(\mathbf{z})$ denotes the induced marginal distribution by $q(\mathbf{x}, \mathbf{y})$ and $q_\phi(\mathbf{z}|\mathbf{x}, \mathbf{y})$. Note here that the relaxation gap is $D_{\mathsf{KL}}(\tilde{q}_\phi(\mathbf{z})\|p_\theta(\mathbf{z}))$, which is again upper bounded by the joint KL divergence (7). Therefore, all relaxation steps in (8) and (11) become tight when the joint distributions induced by data distribution, encoders, priors, and decoders are perfectly consistent with each other, i.e.,

$$q(\mathbf{x}, \mathbf{y})q_\phi(\mathbf{z}|\mathbf{x}, \mathbf{y})q_\phi(\mathbf{u}|\mathbf{z}, \mathbf{x})q_\phi(\mathbf{v}|\mathbf{z}, \mathbf{y}) \equiv p_\theta(\mathbf{z})p_\theta(\mathbf{u})p_\theta(\mathbf{v})p_\theta(\mathbf{x}|\mathbf{z}, \mathbf{u})p_\theta(\mathbf{y}|\mathbf{z}, \mathbf{v}).$$

We note that these relaxation techniques are standard in the literature, the tightness of which deserves a separate future study; for recent related work, refer to Cremer et al. (2018); Poole et al. (2019).

To sum up, the objective function for the joint model (3) is relaxed as

$$\overline{\mathcal{L}}_{\mathbf{xy}\to\mathbf{xy}} + \lambda\overline{I}_q. \tag{12}$$

Note that the additional information regularization with $\lambda > 0$ is only on the common representation $\mathbf{Z}$, but not on $\mathbf{U}, \mathbf{V}$, which we call local representation. By increasing $\lambda > 0$ to a proper degree that does not impedes fitting, we can "reroute" the information flow from $(\mathbf{X}, \mathbf{Y})$ through the common representation $q_\phi(\mathbf{z}|\mathbf{x}, \mathbf{y})$ to the local representations $q_\phi(\mathbf{u}|\mathbf{z}, \mathbf{x}), q_\phi(\mathbf{v}|\mathbf{z}, \mathbf{y})$.

Following similar steps in (7), (8), (9), the objective function in (4) can also be upper bounded as

$$\mathcal{L}_{\mathbf{x}\to\mathbf{x}} \leq \overline{\mathcal{L}}_{\mathbf{x}\to\mathbf{x}} := D_{\mathsf{KL}}(q(\mathbf{x})q_\phi(\mathbf{z}|\mathbf{x})q_\phi(\mathbf{u}|\mathbf{z}, \mathbf{x})\|p_\theta(\mathbf{z})p_\theta(\mathbf{u})p_\theta(\mathbf{x}|\mathbf{z}, \mathbf{u})). \tag{13}$$

For (5), we choose $\mathcal{L}_{\mathbf{xy}\to\mathbf{y}} := D_{\mathsf{KL}}(q(\mathbf{x}, \mathbf{y})q_\phi(\mathbf{z}|\mathbf{x}, \mathbf{y})\|q(\mathbf{x})q_\phi(\mathbf{z}|\mathbf{x})\tilde{p}_\theta(\mathbf{y}|\mathbf{z}))$, which can be viewed as the expected conditional ELBO (Sohn et al., 2015). It can be also subsequently relaxed as

$$\mathcal{L}_{\mathbf{xy}\to\mathbf{y}} \leq \overline{\mathcal{L}}_{\mathbf{xy}\to\mathbf{y}} := D_{\mathsf{KL}}(q(\mathbf{x}, \mathbf{y})q_\phi(\mathbf{z}|\mathbf{x}, \mathbf{y})q_\phi(\mathbf{v}|\mathbf{z}, \mathbf{y})\|q(\mathbf{x})q_\phi(\mathbf{z}|\mathbf{x})p_\theta(\mathbf{v})p_\theta(\mathbf{y}|\mathbf{z}, \mathbf{v})). \tag{14}$$

After all, the final relaxed objective function is given as follows:

$$\overline{\mathcal{L}} := \overline{\mathcal{L}}_{\mathbf{xy}\to\mathbf{xy}} + \lambda\overline{I}_q + \alpha_{\mathbf{x}\to\mathbf{x}}\overline{\mathcal{L}}_{\mathbf{x}\to\mathbf{x}} + \alpha_{\mathbf{y}\to\mathbf{y}}\overline{\mathcal{L}}_{\mathbf{y}\to\mathbf{y}} + \alpha_{\mathbf{xy}\to\mathbf{y}}\overline{\mathcal{L}}_{\mathbf{xy}\to\mathbf{y}} + \alpha_{\mathbf{xy}\to\mathbf{x}}\overline{\mathcal{L}}_{\mathbf{xy}\to\mathbf{x}}. \tag{15}$$

See Figure 2 for an overview of each term in the objective function.

We call the overall framework which consists of all the components in the Wyner model and the additional encoders $q_\phi(\mathbf{z}|\mathbf{x}, \mathbf{y})$, $q_\phi(\mathbf{u}|\mathbf{z}, \mathbf{x})$, $q_\phi(\mathbf{v}|\mathbf{z}, \mathbf{y})$ together with its training objective (15) as *Wyner common representation VAE* or *Wyner VAE* in short. After parameterizing each distribution component in Wyner VAE as standard parametric distributions such as Gaussians, whose parameters are again parameterized by deep neural networks, Wyner VAE can be trained efficiently by the standard reparameterization trick (Kingma and Welling, 2014) as in the standard VAE. (See Appendix B for the Gaussian parameterization and the corresponding objective functions.)

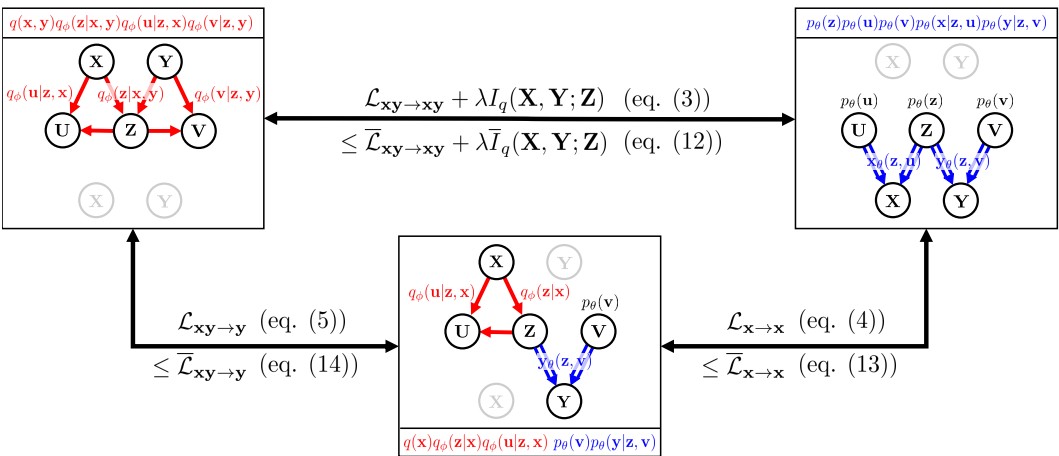

Figure 2: A summary of the training objective for the Wyner VAE (15).

In practice, we can either train the objective function (15) jointly with some positive $\alpha$'s (*joint training*), or train the joint model first letting $\alpha$'s be 0 and then train the marginal encoder in a retrospective manner by freezing the joint model (*two-stage training*) as in Vedantam et al. (2018). The hyperparameter $\alpha$'s may be chosen with cross-validation with validation sets. We elaborate the training schemes used for each experiment in Appendix F.

We remark the role of the encoders $q_\phi(\mathbf{z}|\mathbf{x}, \mathbf{y})$, $q_\phi(\mathbf{u}|\mathbf{z}, \mathbf{x})$, $q_\phi(\mathbf{v}|\mathbf{z}, \mathbf{y})$. The joint encoder $q_\phi(\mathbf{z}|\mathbf{x}, \mathbf{y})$ appears in the optimization problem (1) and plays as a *reference distribution* in learning the components of the Wyner model. The variational encoders $q_\phi(\mathbf{u}|\mathbf{z}, \mathbf{x})$ and $q_\phi(\mathbf{v}|\mathbf{z}, \mathbf{y})$ are introduced to remove the intractable induced distributions $\tilde{p}_\theta(\mathbf{z}|\mathbf{x})$ and $\tilde{p}_\theta(\mathbf{z}|\mathbf{y})$, satisfying the correct conditional independence structure implied by the decoder model $p_\theta(\mathbf{z})p_\theta(\mathbf{u})p_\theta(\mathbf{v})p_\theta(\mathbf{x}|\mathbf{z}, \mathbf{u})p_\theta(\mathbf{y}|\mathbf{z}, \mathbf{v})$, that is, $q_\phi(\mathbf{z}, \mathbf{u}, \mathbf{v}|\mathbf{x}, \mathbf{y}) = q_\phi(\mathbf{z}|\mathbf{x}, \mathbf{y})q_\phi(\mathbf{u}|\mathbf{z}, \mathbf{x})q_\phi(\mathbf{v}|\mathbf{z}, \mathbf{y})$, or equivalently $\mathbf{U} \perp\!\!\!\perp (\mathbf{V}, \mathbf{Y})|\mathbf{Z}$ and $\mathbf{V} \perp\!\!\!\perp (\mathbf{U}, \mathbf{X})|\mathbf{Z}$. If we learn a succinct common representation $q_\phi(\mathbf{z}|\mathbf{x}, \mathbf{y})$ (e.g., a shared concept) from $(\mathbf{X}, \mathbf{Y})$, then $q_\phi(\mathbf{u}|\mathbf{z}, \mathbf{x})$ would capture the remaining randomness $\mathbf{U}$ of $\mathbf{X}$ (e.g., texture and style). We call this decomposition of the pair $(\mathbf{X}, \mathbf{Y})$ into the common representation $\mathbf{Z}$ and the local representation $\mathbf{U}, \mathbf{V}$ as *common-local information decomposition* of $(\mathbf{X}, \mathbf{Y})$. We refer to $(\mathbf{Z}, \mathbf{U}, \mathbf{V})$ as the *joint representation* of $(\mathbf{X}, \mathbf{Y})$, to distinguish it from the common representation $\mathbf{Z}$. Provided that Wyner VAE achieves a good information decomposition, the variational encoders then can be used to explicitly in finding the local representations $\mathbf{U}$ and $\mathbf{V}$ from the data variables $\mathbf{X}$ and $\mathbf{Y}$.

### 2.4 SAMPLING WITH STYLE CONTROL

As alluded to above, the variational encoders $q_\phi(\mathbf{u}|\mathbf{z}, \mathbf{x}), q_\phi(\mathbf{v}|\mathbf{z}, \mathbf{y})$ can be used in sampling tasks *with style control* as a local representation (i.e., style) extractor. We illustrate how to perform conditional sampling with style control (Fig. 3 (e)). Suppose that $(\mathbf{X}, \mathbf{Y})$ is a pair of correlated images generated from the common concept but from different domains. Given an image $\mathbf{y}_0$, we can extract the style information $\mathbf{V}_0$ from $\mathbf{y}_0$ by sampling $(\mathbf{Z}_0, \mathbf{V}_0)$ from $q_\phi(\mathbf{z}|\mathbf{y})q_\phi(\mathbf{v}|\mathbf{z}, \mathbf{y})$ (Fig. 3 (d)). We then generate $\mathbf{Y}_j$ from an image $\mathbf{x}_j$ similar to conditional sampling (Fig. 3 (c)), while replacing the randomly drawn local representation $\mathbf{V} \sim p_\theta(\mathbf{v})$ with the previously extracted style $\mathbf{V}_0$, thereby the generated images $\mathbf{Y}_{0,j}$'s are of the same style as the reference image $\mathbf{y}_0$. In a similar manner, we can also perform joint sampling with a fixed style given a style reference data pair $(\mathbf{x}_0, \mathbf{y}_0)$, by mixing a randomly drawn common representation $\mathbf{Z}$ from the prior $p_\theta(\mathbf{z})$ with the extracted style variables $(\mathbf{u}_0, \mathbf{v}_0)$.

### 2.5 DEGENERATE CASES IN WYNER VAE

Two degenerate cases may arise in Wyner VAE. The first case is where the common variable $\mathbf{Z}$ captures all information of $(\mathbf{X}, \mathbf{Y})$, while $\mathbf{U}$ and $\mathbf{V}$ capture none. Joint VAE (JVAE) (Vedantam et al., 2018) and joint multimodal VAE (JMVAE) (Suzuki et al., 2016) inherently assume this degenerate case, as it is discssued in the next section. Wyner VAE is able to avoid such degeneracy by explicitly

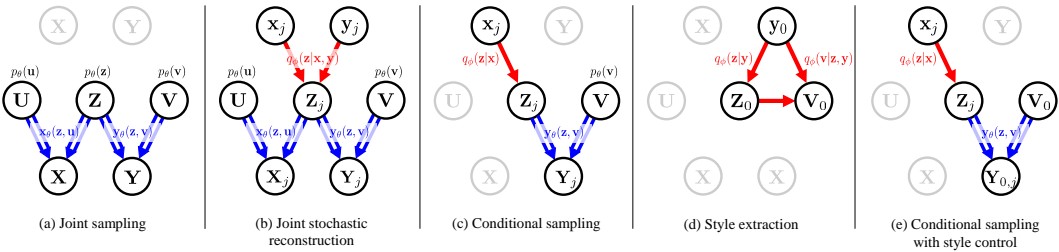

Figure 3: Schematics for selected sampling tasks. Double arrows denote deterministic mapping.

Table 1: Summary of related work. (J: joint generation, C: conditional generation, S: style control.)

|  | C | J | S |
|---|---|---|---|
| JVAE (Vedantam et al., 2018), JMVAE (Suzuki et al., 2016) | O | O | X |
| CVAE (Sohn et al., 2015) | O | X | O |
| VCCA-private (Wang et al., 2016) | O | O | O |
| VIB (Alemi et al., 2017) | O | X | X |
| **Wyner VAE** | O | O | O |

having the local variables and controlling the common regularization parameter $\lambda > 0$. On the other extreme, $\mathbf{Z}$ may capture no information, while $\mathbf{U}$ and $\mathbf{V}$ capture all the information of $\mathbf{X}$ and $\mathbf{Y}$, respectively. It may happen in Wyner VAE if the regularization parameter $\lambda$ is too large. To avoid the degeneracy, we need to choose a proper $\lambda$ by cross-validation.

## 3 RELATED WORK

In this section, we compare the proposed Wyner VAE to the existing models, deferring a detailed description of the encoder, decoder, prior components and objective functions of each model to Appendix C. We provide a summary for capabilities of each model in Table 1.

**VAE models.** Wyner VAE can be viewed as a generalization of the probabilistic model (i.e., encoder and decoder) assumed in two existing joint VAEs— JVAE (Vedantam et al., 2018) and JMVAE (Suzuki et al., 2016)—as alluded in the previous paragraph. These models implement a similar idea of performing joint and conditional generation tasks via a symmetric Markov chain $\mathbf{X} \to \mathbf{W} \to \mathbf{Y}$, where $\mathbf{W}$ is the *joint representation* of $(\mathbf{X}, \mathbf{Y})$. In other words, these models can be derived by removing the local variables $\mathbf{U}$ and $\mathbf{V}$ in Wyner VAE. In Section 4, we demonstrated that the local variables in Wyner VAE help generate a variety of samples compared to JVAE.

The same decoder structure of Wyner VAE with the "shared" ($\mathbf{Z}$) and the "private" ($\mathbf{U}, \mathbf{V}$) latent variables has been also studied in the context of multi-view learning (Shon et al., 2006; Ek et al., 2008; Salzmann et al., 2010; Damianou et al., 2012) mostly based on a linear analysis such as canonical correlation analysis (CCA). More recently, variational CCA-private (VCCA-private) (Wang et al., 2016) was proposed to learn the decoder model with variational encoders $q_\phi(\mathbf{z}|\mathbf{x})$, $q_\phi(\mathbf{u}|\mathbf{x})$, and $q_\phi(\mathbf{v}|\mathbf{y})$, with the encoder model $q_\phi(\mathbf{z}, \mathbf{u}, \mathbf{v}|\mathbf{x}, \mathbf{y}) = q_\phi(\mathbf{z}|\mathbf{x})q_\phi(\mathbf{u}|\mathbf{x})q_\phi(\mathbf{v}|\mathbf{y})$ to directly capture the conditional model from $\mathbf{X}$ to $\mathbf{Z}$ to $\mathbf{Y}$. On the other hand, Wyner VAE relies on the conditional independence structure $q_\phi(\mathbf{z}, \mathbf{u}, \mathbf{v}|\mathbf{x}, \mathbf{y}) = q_\phi(\mathbf{z}|\mathbf{x}, \mathbf{y})q_\phi(\mathbf{u}|\mathbf{z}, \mathbf{x})q_\phi(\mathbf{v}|\mathbf{z}, \mathbf{y})$, which is naturally induced by the decoder model. We argue that this choice of encoder model in Wyner VAE may capture better semantic meaning of the local (private) random variables $\mathbf{U}$ and $\mathbf{V}$, thereby leading a better generative performance; see, e.g., Fig. 6.

Conditional VAE (CVAE) (Sohn et al., 2015) directly models the conditional distribution $q(\mathbf{y}|\mathbf{x})$, obtained by simply conditioning every component in the vanilla VAE for $q(\mathbf{y})$ with the conditioning variable $\mathbf{X}$. If $\mathbf{Y}$ is an image and $\mathbf{X}$ is an attribute of the image, a latent representation $\mathbf{V}$ in CVAE needs to capture the redundant information of $\mathbf{Y}$, which is not contained in $\mathbf{X}$, i.e., style information of $\mathbf{Y}$ given $\mathbf{X}$. Wyner VAE can be viewed as a combination of two CVAEs with $\mathbf{Z}$ as a common conditioning variable, being capable of bidirectional sampling in its nature. Yet, if $\mathbf{X}$ is high-dimensional, the conditional models like CVAE in general tend to overfit the input data of

Table 2: Wyner model vs. the IB principle (Tishby et al., 1999).

|  | **Wyner model** | **IB principle** |
|---|---|---|
| Motivating problem | Channel synthesis, distributed simulation | Lossy compression |
| Probabilistic model | $\mathbf{X} \to \mathbf{Z} \to \mathbf{Y}$ | $\mathbf{Z} \to \mathbf{X} \to \mathbf{Y}$ |
| Direction of inference | Bidirectional | Unidirectional |
| Measure of succinctness | $I(\mathbf{X}, \mathbf{Y}; \mathbf{Z})$ | $I(\mathbf{X}; \mathbf{Z})$ |
| Measure of fit/relevance | $D(p, q)$ | $I(\mathbf{Y}; \mathbf{Z})$ |
| Optimal quantity | $J(\mathbf{X}; \mathbf{Y})$ | N/A |

$\mathbf{X}$ (Dutordoir et al., 2018). To address this problem, a subsequent related work, bottleneck conditional density estimation (BCDE) (Shu et al., 2017), proposed to learn joint and conditional VAE models simultaneously by softly tying the parameters of the two models for regularization. We note that Wyner VAE naturally addresses such problem by using a unified single probabilistic model for both joint and conditional distribution learning, finding a succinct common representation $\mathbf{Z}$ for regularization.

**Information bottleneck principle.** The information bottleneck (IB) principle (or method) (Tishby et al., 1999) is a widely known information theoretic approach in representation learning especially for *discriminative tasks*, i.e., when the target variable $\mathbf{Y}$ is a function of $\mathbf{X}$ and/or even discrete. Motivated by lossy compression, the IB principle proposes to find a *compressed* representation $\mathbf{Z}$ from the input variable $\mathbf{X}$ (i.e., $q_\phi(\mathbf{z}|\mathbf{x})$) while maximizing the relevance of $\mathbf{Z}$ in *predicting* the target variable $\mathbf{Y}$ as the minimizer of the optimization problem $\text{minimize}_{q_\phi(\mathbf{z}|\mathbf{x})} \beta I_q(\mathbf{X}; \mathbf{Z}) - I_q(\mathbf{Y}; \mathbf{Z})$, where $(\mathbf{X}, \mathbf{Y}, \mathbf{Z}) \sim q(\mathbf{x}, \mathbf{y})q_\phi(\mathbf{z}|\mathbf{x})$.

The foremost difference between the IB principle and our approach is in the underlying Markov chains: our symmetric Markov assumption $\mathbf{X} \to \mathbf{Z} \to \mathbf{Y}$ is more natural than $\mathbf{Z} \to \mathbf{X} \to \mathbf{Y}$ of IB, when guessing $\mathbf{Y}$ based on $\mathbf{Z}$ as a representation of $\mathbf{X}$. Further, our framework aims to find a certain common information structure characterized by Wyner's common information with proper analogies to generation tasks of our interest, whereas the compressed-from-$\mathbf{X}$ yet relevant-to-$\mathbf{Y}$ representation $\mathbf{Z}$ in the IB principle lacks its operational meaning, relying on a rather weak analogy to lossy compression. We summarize other differences in various aspects in Table 2. In particular, we compare Wyner VAE with variational IB (VIB) (Alemi et al., 2017) in the experiments, which is an instantiation of the IB principle based on a variational technique that can be implemented with neural networks. As empirically shown below, VIB is not suitable for conditional generative tasks if the target variable $\mathbf{Y}$ is high-dimensional.

## 4 EXPERIMENTS

We empirically demonstrate that Wyner VAE outperforms JVAE, CVAE, VCCA-private, and VIB, for joint/conditional generation tasks and style manipulation on various datasets. We defer the implementation details and training schemes used for each experiment to Appendix F.

**Synthetic data.** We first performed an experiment with a mixture of Gaussians (MoG) dataset as a toy example. We considered a pair of 10-dim. MoG random vectors $(\mathbf{X}, \mathbf{Y})$ only correlated through a label $\mathbf{Z} \sim \text{Unif}([1, 2, 3, 4, 5])$ (common information) and 5-dim. Gaussian random vectors $\mathbf{U}, \mathbf{V} \sim \mathcal{N}(0, I_5)$ (local randomness in each variable). We used the Gaussian latent variables $(\mathbf{Z}, \mathbf{U}, \mathbf{V})$ of dimensions $(10, 10, 10)$, trained each model for 500 epochs (separate 50 epochs for each marginal encoder for JVAE and Wyner VAE) with a training data of size 50k, and summarized the numerical results in Fig. 4 and Table 3, which were evaluated with a test data of size 10k. See Fig. 9 in Appendix E.1 for some visualizations.

In Fig. 4 (a,b), Wyner VAE with $\lambda = 0$ performed best for fitting joint distributions, but did not excel in conditional log-likelihoods. We observe that the performance of Wyner VAE on the test data gets improved throughout training by increasing $\lambda$: $\lambda = 0.05$ achieved a good conditional performance without too much sacrifice in the joint performance, while a larger value of $\lambda(= 0.1)$ interfered fitting to the distribution, failing to capture the essential common information structure. Overall, Wyner VAE with $\lambda$ control outperformed the other models. CVAE tends to overfit quickly as noted earlier.

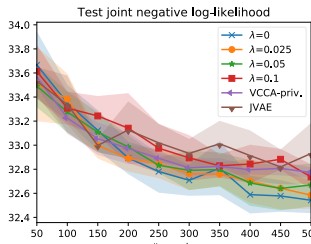 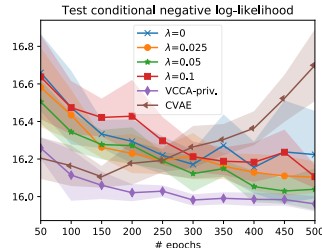 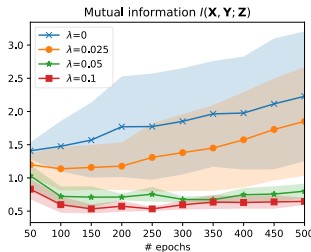

Figure 4: Numerical evaluations for the MoG experiment. For each point of the plots, we trained 10 different models and plotted average values with the shaded region that shows the standard deviation. (Two largest and smallest outliers were dropped for each point.)

Table 3: Best negative log-likelihood (nll) values during 500 epochs of training for MoG dataset.

| | Joint nll | Conditional nll |
|---|---|---|
| JVAE (Vedantam et al., 2018) | $32.82 \pm 0.14$ | $32.80 \pm 0.76$ |
| Wyner VAE ($\lambda = 0$) | $\mathbf{32.54 \pm 0.11}$ | $16.16 \pm 0.14$ |
| Wyner VAE ($\lambda = 0.05$) | $\mathbf{32.64 \pm 0.18}$ | $16.03 \pm 0.07$ |
| VCCA-private (Wang et al., 2016) | $32.77 \pm 0.04$ | $\mathbf{15.96 \pm 0.04}$ |
| CVAE (Sohn et al., 2015) | - | $16.11 \pm 0.06$ |
| VIB (Alemi et al., 2017) ($\beta = 0.1$) | - | $503.91 \pm 2.79$ |

JVAE and VIB performed extremely worse in test conditional log-likelihoods compared to others as in Table 3. JVAE failed to capture the common information structure as the training epochs increased, while VIB was only able to capture the average behaviors — we demonstrate how these models failed in this toy dataset in Appendix E.1. Although VCCA-private achieved the best conditional likelihood with a comparable joint likelihood performance, this model fails to learn more complex distributions as illustrated in Fig. 6 and Appendix E.2, E.3.

Fig. 4 (c) shows that $\lambda$ can control the common mutual information $I_q(\mathbf{X}, \mathbf{Y}; \mathbf{Z})$ in Wyner VAE — in particular, $\lambda \in \{0.05, 0.1\}$ kept the (estimated) mutual information at a constant level. We remark, however, that the MoG dataset has $J(\mathbf{X}; \mathbf{Y}) = \log 5$, while the mutual information estimates are not around the true value (Fig. 4). We attribute this mismatch to modeling the underlying discrete latent variable with a continuous, Gaussian vector. Note that we deliberately used the standard Gaussian model assuming that we do not have a prior knowledge on the dataset generating process.

**MNIST and SVHN datasets.** We performed experiments with image datasets MNIST (Le-Cun, 1998) and SVHN (Netzer et al., 2011), by randomly pairing digit images only through their labels. In particular, we constructed two dataset, MNIST–MNIST add-1 dataset, where $\text{label}(\mathbf{Y}_i) = \text{label}(\mathbf{X}_i) + 1 \pmod{10}$, and MNIST–SVHN dataset, where $\text{label}(\mathbf{Y}_i) = \text{label}(\mathbf{X}_i)$.

For MNIST–MNIST add-1 dataset, we trained Wyner VAE with different choices of $\lambda \in \{0.0, 0.1, \dots, 0.5\}$ to show the effect of $\lambda$ on the generative performance of Wyner VAE. Fig. 5 (a) corroborates our main claim that $\lambda > 0$ helps learning succinct representation in terms of small $I_q(\mathbf{X}, \mathbf{Y}; \mathbf{Z})$ and there exists a sweet spot ($\lambda = 0.1$) that strikes the balance between the fitting and the succinctness.

We also evaluated the label accuracy using the pre-trained Le-Net5 (LeCun et al., 1998) classification network of accuracy 99.1%, and the per-pixel variance of samples from conditional generation. Note that both high accuracy and high variance are desired for good generative models. Fig. 5 (b) shows that by rerouting the information flow through $\mathbf{Z}$ to $\mathbf{U}, \mathbf{V}$, $\lambda > 0$ helps $\mathbf{U}, \mathbf{V}$ capture the style of images with a small sacrifice in label accuracy.

We present image samples to visualize the effect of $\lambda$ in Wyner VAE and the superiority of Wyner VAE over the existing models. We performed four sampling tasks—conditional generation, conditional generation with style control, joint stochastic generation, and joint sampling with style control; see Fig. 3—for both MNIST–MNIST add-1 and MNIST–SVHN datasets, but here we present only a few

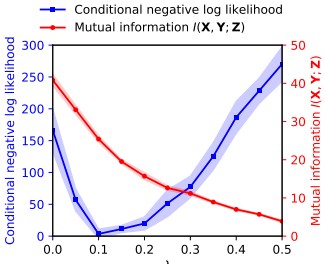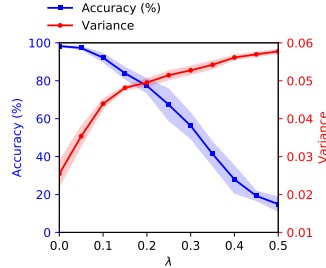

Figure 5: Numerical evaluations of Wyner VAE for conditional generation of MNIST–MNIST add-1 dataset. The plots were generated similarly as Fig. 4. See also Table 5 in the Appendix.

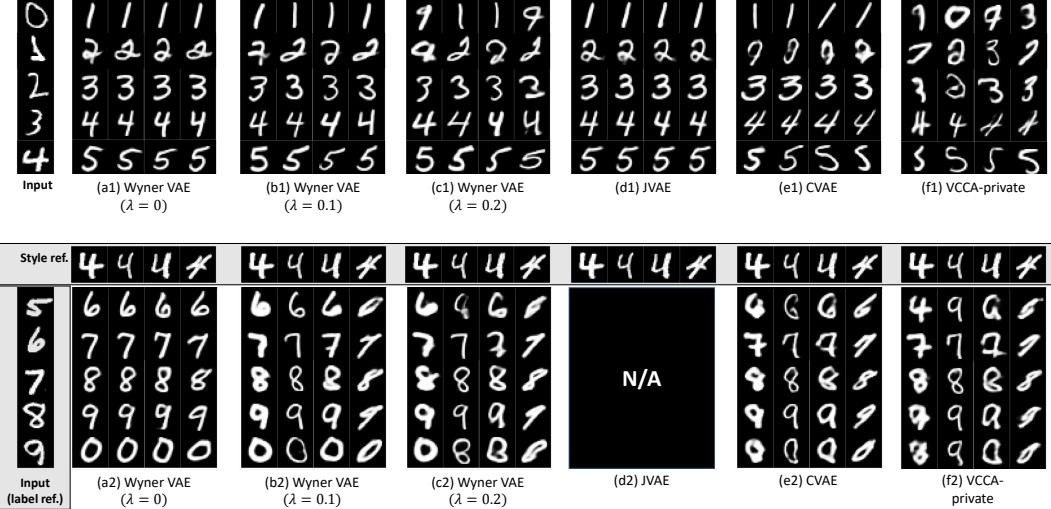

Figure 6: Samples from Wyner VAE and the other models for MNIST–MNIST add-1 dataset. (a1-f1) Conditional sampling. (a2-f2) Conditional sampling with style control. For both tasks, the leftmost column denotes the conditioning input to the models.

samples for illustration. We refer the interested reader to Appendix E.2, E.3 for a full comparison. We defer all the samples from VIB to the appendix, as it only generated same "average" images.

Fig. 6 presents samples of MNIST–MNIST pairs by conditional generation with and without style control. Fig. 6 (a1-c1) and Fig. 6 (a2-c2) demonstrated how the variations in the generated images and the style information captured in $\mathbf{V}$ are affected by varying $\lambda$, respectively. (We illustrate how we performed conditional generation with style control for CVAE and VCCA-private in Appendix C.) $\lambda = 0.1$ showed the best conditional generation results, consistent with Fig. 5 (a). JVAE generated images without much variation. VCCA-private erred frequently in guessing the labels, which implies that the shared representation $\mathbf{Z}$ in VCCA-private does not capture the "common information".

Fig. 7 presents samples of MNIST–SVHN pair from conditional sampling with style control (Fig. 7 (a,b)) and two variations of joint sampling tasks (Fig. 7 (c-f)). Fig. 7 (c,d) show joint stochastic reconstruction: $\mathbf{Z}$ is inferred from the label reference data, and samples are generated jointly by drawing local randomness $(\mathbf{U}, \mathbf{V}) \sim p_\theta(\mathbf{u})p_\theta(\mathbf{v})$; see Fig. 3 (b). Fig. 7 (e,f) show joint sampling with style control: similarly to conditional sampling with style control, we generated joint samples by drawing common randomness $\mathbf{Z} \sim p_\theta(\mathbf{z})$ with a specified local information from the style reference. In all cases, we observe that $\lambda = 0.1$ achieves better style manipulation over $\lambda = 0$, indicating that $\lambda > 0$ helps separating style information from common information.

**MNIST quadrant prediction dataset.** We performed a quadrant prediction task (Sohn et al., 2015) with a static, binary MNIST dataset (Larochelle and Murray, 2011), using the Bernoulli observation model for decoders. Specifically, we split each digit image into two parts into left ($\mathbf{X}$; conditioning)

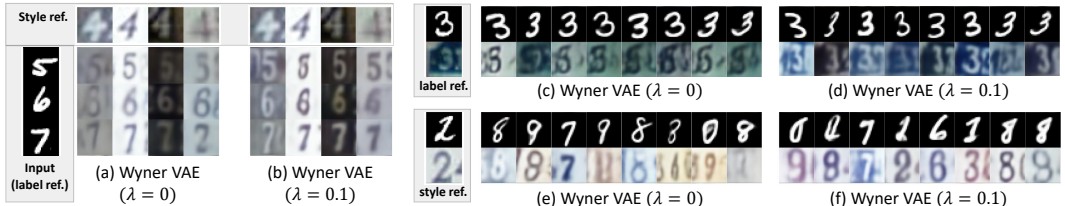

Figure 7: Samples from Wyner VAE for MNIST–SVHN dataset. (a,b) Conditional generation with style control. (c,d) Joint stochastic reconstruction. (e,f) Joint generation with style control. $\lambda = 0.1$ helps local latent variables capture style information, and the generated samples exhibit the effect compared to $\lambda = 0$.

Table 4: Best nll values during 1000 epochs of training for MNIST left-right prediction task. The conditional nll value for CVAE is taken from Sohn et al. (2015).

|                              | Joint nll         | Conditional nll    |
|------------------------------|-------------------|--------------------|
| JVAE (Vedantam et al., 2018) | $91.05 \pm 0.03$  | $45.22 \pm 0.03$   |
| Wyner VAE ($\lambda = 0$)    | $\mathbf{87.74 \pm 0.05}$ | $\mathbf{43.87 \pm 0.03}$ |
| Wyner VAE ($\lambda = 0.15$) | $89.66 \pm 0.03$  | $\mathbf{43.85 \pm 0.03}$ |
| CVAE (Sohn et al., 2015)     | -                 | $44.73$            |

and right ($\mathbf{Y}$; target). In this case, the most succinct common representation $\mathbf{Z}$ is a nontrivial object in contrast to a label (or its bijection) in the previous experiments. Table 4 summarizes negative log-likelihoods for Wyner VAE, JVAE, and CVAE on the test dataset. Note that Wyner VAE with with $\lambda = 0$ performs best, while an additional regularization parameter $\lambda = 0.15$ only gives a marginal improvement in conditional performance. For this specific case, having the local variables $\mathbf{U}$ and $\mathbf{V}$ is enough to capture the local information, as the underlying data seems to have much complex common information structure compared to a discrete label in the previous cases.

**CelebA dataset.** We performed an experiment with CelebA dataset (Liu et al., 2015), which is a degenerate case in the sense that $\mathbf{X}$ is a function of $\mathbf{Y}$. While any bijection of $\mathbf{X}$ is common representation $\mathbf{Z}$ attaining $J(\mathbf{X}; \mathbf{Y})$ in this case, we demonstrated some merits of using Wyner VAE over CVAE in terms of better common representation learning in Appendix E.4.

## 5 CONCLUDING REMARKS

Cuff's channel synthesis and Wyner's distributed simulation are another manifestation of Occam's razor by finding the simplest probabilistic structure that connects one random object to another. The proposed Wyner VAE finds this succinct structure in a disciplined yet efficient manner, and provides a theoretically sound alternative to the information bottleneck principle. The experimental results demonstrated the potential of our approach as a new way of learning joint and conditional generation tasks with optimal representation learning that can be further developed and refined for more complex dataset such as auditory, text, or a pair of those.

A few remaining questions from experiments are in order. First, this paper does not address how close the estimated common information from a learned Wyner VAE model is to the true Wyner's common information $J(\mathbf{X}; \mathbf{Y})$, and it may be interesting to devise a better estimation technique of Wyner's common information from data. We emphasize that, however, the main goal of this paper is to demonstrate the advantages of the probabilistic structure of Wyner VAE and the regularization parameter $\lambda$ in various joint and conditional generation tasks. Second, it would be interesting to investigate the relation between the variational approximation gaps and the quality of learned representations in Wyner VAE (Cremer et al., 2018; Poole et al., 2019).

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

## A  A QUICK OVERVIEW ON VARIATIONAL AUTOENCODERS

Variational autoencoder (VAE) (Kingma and Welling, 2014; Rezende et al., 2014) is a class of deep generative models that aim to *simulate* the unknown distribution $q(\mathbf{x})$ underlying the data $\mathbf{x}_1, \ldots, \mathbf{x}_N$ to generate new samples from this distribution efficiently. Let $q_{\text{emp}}(\mathbf{x})$ be the empirical distribution defined by the sample. Assume a generative latent variable model $p_\theta(\mathbf{z})p_\theta(\mathbf{x}|\mathbf{z})$ to model the underlying distribution $q(\mathbf{x})$. One of the standard approach to learn each component in the model, the prior $p_\theta(\mathbf{z})$ and the decoder $p_\theta(\mathbf{x}|\mathbf{z})$, is the maximum likelihood approach that aims to solve

$$\underset{\theta}{\text{maximize}} \sum_{i=1}^{N} \log \tilde{p}_\theta(\mathbf{x}_i), \tag{16}$$

or equivalently,

$$\underset{\theta}{\text{minimize}} \ D_{\text{KL}}(q_{\text{emp}}(\mathbf{x}) \| \tilde{p}_\theta(\mathbf{x})), \tag{17}$$

where $\tilde{p}_\theta(\mathbf{x})$ is the induced distribution characterized by $p_\theta(\mathbf{z})p_\theta(\mathbf{x}|\mathbf{z})$, i.e., $\tilde{p}_\theta(\mathbf{x}) := \int p_\theta(\mathbf{z})p_\theta(\mathbf{x}|\mathbf{z}) \, d\mathbf{z}$. However, it is often computationally hard to solve the optimization problem directly due to the induced distribution $\tilde{p}_\theta(\mathbf{x})$ that involves an integration over a high-dimensional space.

In variational Bayesian learning approach (see, e.g., (Blei et al., 2017)), an approximate posterior $q_\phi(\mathbf{z}|\mathbf{x})$ (also called as an encoder) is introduced to relax the objective (16). Here we present a short derivation of the well-known VAE objective function. Note that the objective in (16) can be upper bounded as

$$D_{\text{KL}}(q_{\text{emp}}(\mathbf{x}) \| \tilde{p}_\theta(\mathbf{x})) \leq D_{\text{KL}}(q_{\text{emp}}(\mathbf{x}) \| \tilde{p}_\theta(\mathbf{x})) + \mathbb{E}_{q_{\text{emp}}(\mathbf{x})}[D_{\text{KL}}(q_\phi(\mathbf{z}|\mathbf{X}) \| \tilde{p}_\theta(\mathbf{z}|\mathbf{X}))] \tag{18}$$

$$= D_{\text{KL}}(q_{\text{emp}}(\mathbf{x})q_\phi(\mathbf{z}|\mathbf{x}) \| \tilde{p}_\theta(\mathbf{x})\tilde{p}_\theta(\mathbf{z}|\mathbf{x})) \tag{19}$$

$$= D_{\text{KL}}(q_{\text{emp}}(\mathbf{x})q_\phi(\mathbf{z}|\mathbf{x}) \| p_\theta(\mathbf{z})p_\theta(\mathbf{x}|\mathbf{z})), \tag{20}$$

where $\tilde{p}_\theta(\mathbf{z}|\mathbf{x})$ is the induced posterior characterized by $p_\theta(\mathbf{z})p_\theta(\mathbf{x}|\mathbf{z})$. (18) follows from the non-negativity of the KL divergence, and (19) follows from the chain rule of the KL divergence (see, e.g, Cover and Thomas (2006)). Note that the final relaxed form (20) does not contain the intractable term $\tilde{p}_\theta(\mathbf{x})$. The variational relaxation is tight if and only if $D_{\text{KL}}(q_\phi(\mathbf{z}|\mathbf{x}) \| p_\theta(\mathbf{z}|\mathbf{x})) = 0$ for all $\mathbf{x}$.

The upper bound (20) is the objective function for the standard VAE model (Kingma and Welling, 2014; Rezende et al., 2014). That is, the standard VAE model aims to solve the following optimization problem:

$$\underset{\theta,\phi}{\text{minimize}} \ D_{\text{KL}}(q_{\text{emp}}(\mathbf{x})q_\phi(\mathbf{z}|\mathbf{x}) \| p_\theta(\mathbf{z})p_\theta(\mathbf{x}|\mathbf{z})). \tag{21}$$

To express the objective function in a more standard form in the literature, we add a constant $h(q_{\text{emp}}(\mathbf{x}))$, the differential entropy of $q_{\text{emp}}(\mathbf{x})$, and then derive

$$D_{\text{KL}}(q_{\text{emp}}(\mathbf{x})q_\phi(\mathbf{z}|\mathbf{x}) \| p_\theta(\mathbf{z})p_\theta(\mathbf{x}|\mathbf{z})) + h(q_{\text{emp}}(\mathbf{x})) \tag{22}$$

$$= \mathbb{E}_{q_{\text{emp}}(\mathbf{x})} \left[ D_{\text{KL}}(q_\phi(\mathbf{z}|\mathbf{X}) \| p_\theta(\mathbf{z})) + \int q_\phi(\mathbf{z}|\mathbf{X}) \log \frac{1}{p_\theta(\mathbf{X}|\mathbf{z})} \, d\mathbf{z} \right] =: \mathbb{E}_{q_{\text{emp}}(\mathbf{x})} \left[ \mathcal{L}_{\theta,\phi}(\mathbf{X}) \right]. \tag{23}$$

The loss function $\mathcal{L}_{\theta,\phi}(\mathbf{x})$ in (23) with negation is called the *evidence lower bound* (ELBO) in the literature, since $-\mathcal{L}_{\theta,\phi}(\mathbf{x})$ lower bounds the *evidence* $\log p_\theta(\mathbf{x})$ for each $\mathbf{x}$. The KL divergence term and the expected log loss term are called as the *regularization* term and the *reconstruction* term, respectively.

Assume that the observed variable $\mathbf{X}$ is continuous for simplicity. The most standard parameterization of the components in the VAE model is the diagonal Gaussian parameterization

$$p_\theta(\mathbf{z}) = \mathcal{N}(\mathbf{z}|0, \text{diag}(\boldsymbol{\sigma}_{0,\theta}^2)), \tag{24}$$

$$p_\theta(\mathbf{x}|\mathbf{z}) = \mathcal{N}(\mathbf{x}|\mathbf{x}_\theta(\mathbf{z}), \text{diag}(\boldsymbol{\sigma}_\theta^2(\mathbf{z}))), \tag{25}$$

$$q_\phi(\mathbf{z}|\mathbf{x}) = \mathcal{N}(\mathbf{z}|\mathbf{z}_\phi(\mathbf{x}), \text{diag}(\boldsymbol{\sigma}_\phi^2(\mathbf{x}))), \tag{26}$$

where each function may be parameterized by a neural network. Here, $\boldsymbol{\sigma}^2$ denotes a vector of a proper dimension and $\text{diag}(\boldsymbol{\sigma}^2)$ denotes a diagonal matrix with diagonal entries $\boldsymbol{\sigma}^2$. Often, the covariance

of the prior is taken to be isotropic and constant such as $\boldsymbol{\sigma}_{0,\theta}^2 = \mathbb{1}$, but $\boldsymbol{\sigma}_{0,\theta}^2$ can also be set as an independent trainable parameter as in this work. Note that the diagonal Gaussian parameterization $p_\theta(\mathbf{x}|\mathbf{z})$ is a formal modeling assumption to have a tractable density, which is required to evaluate the log loss $\log 1/p_\theta(\mathbf{x}|\mathbf{z})$ in the reconstruction loss. This formal Gaussian noise also plays a role in estimating likelihoods. However, after training, the decoder variance $\boldsymbol{\sigma}_\theta^2(\mathbf{z})$ is dropped and the resulting decoder is used as deterministic: $\mathbf{z} \mapsto \mathbf{x}_\theta(\mathbf{z})$.

With this parameterization, the loss function $\mathcal{L}_{\theta,\phi}(\mathbf{x})$ in (23) can be estimate efficiently for each $\mathbf{x}$ via a Monte Carlo approximation by sampling $\mathbf{z} \sim q_\phi(\mathbf{z}|\mathbf{x})$. The overall objective function then can be minimized with a gradient based optimization algorithm like Adam (Kingma and Ba, 2014) based on the reparameterization trick (Kingma and Welling, 2014; Rezende and Mohamed, 2015) and backpropagation.

# B  STANDARD IMPLEMENTATION OF WYNER VAE

## B.1  GAUSSIAN PARAMETERIZATION

As elaborated in Appendix A on VAEs, we use a standard Gaussian parameterization for Wyner VAE in our experiments. Concretely, we let

$$p_\theta(\mathbf{z}) = \mathcal{N}(\mathbf{z}|0, \operatorname{diag}(\boldsymbol{\sigma}_{0,\theta}^2)), \tag{27}$$

$$p_\theta(\mathbf{u}) = \mathcal{N}(\mathbf{u}|0, \operatorname{diag}(\boldsymbol{\sigma}_{1,\theta}^2)), \tag{28}$$

$$p_\theta(\mathbf{v}) = \mathcal{N}(\mathbf{v}|0, \operatorname{diag}(\boldsymbol{\sigma}_{2,\theta}^2)), \tag{29}$$

$$p_\theta(\mathbf{x}|\mathbf{z}, \mathbf{u}) = \mathcal{N}(\mathbf{x}|\mathbf{x}_\theta(\mathbf{z}, \mathbf{u}), \operatorname{diag}(\boldsymbol{\sigma}_\theta^2(\mathbf{z}, \mathbf{u}))), \tag{30}$$

$$p_\theta(\mathbf{y}|\mathbf{z}, \mathbf{v}) = \mathcal{N}(\mathbf{y}|\mathbf{y}_\theta(\mathbf{z}, \mathbf{v}), \operatorname{diag}(\boldsymbol{\sigma}_\theta^2(\mathbf{z}, \mathbf{v}))), \tag{31}$$

$$q_\phi(\mathbf{z}|\mathbf{x}, \mathbf{y}) = \mathcal{N}(\mathbf{z}|\mathbf{z}_{012,\phi}(\mathbf{x}, \mathbf{y}), \operatorname{diag}(\boldsymbol{\sigma}_{012,\phi}^2(\mathbf{x}, \mathbf{y}))), \tag{32}$$

$$q_\phi(\mathbf{u}|\mathbf{z}, \mathbf{x}) = \mathcal{N}(\mathbf{u}|\mathbf{u}_\phi(\mathbf{z}, \mathbf{x}), \operatorname{diag}(\boldsymbol{\sigma}_{1,\phi}^2(\mathbf{z}, \mathbf{x}))), \tag{33}$$

$$q_\phi(\mathbf{v}|\mathbf{z}, \mathbf{y}) = \mathcal{N}(\mathbf{v}|\mathbf{v}_\phi(\mathbf{z}, \mathbf{y}), \operatorname{diag}(\boldsymbol{\sigma}_{2,\phi}^2(\mathbf{z}, \mathbf{y}))), \tag{34}$$

$$q_\phi(\mathbf{z}|\mathbf{x}) = \mathcal{N}(\mathbf{z}|\mathbf{z}_{01,\phi}(\mathbf{x}), \operatorname{diag}(\boldsymbol{\sigma}_{01,\phi}^2(\mathbf{x}))), \tag{35}$$

$$q_\phi(\mathbf{z}|\mathbf{y}) = \mathcal{N}(\mathbf{z}|\mathbf{z}_{02,\phi}(\mathbf{y}), \operatorname{diag}(\boldsymbol{\sigma}_{01,\phi}^2(\mathbf{y}))), \tag{36}$$

where each function may be parameterized by a neural network.

## B.2  OBJECTIVE FUNCTIONS

We can rewrite the objective function $\overline{\mathcal{L}}_{\mathbf{xy} \to \mathbf{xy}}$ in (12) in terms of the expectation of ELBO as in (23):

$$D_{\mathsf{KL}}(q_\phi(\mathbf{x}, \mathbf{y}, \mathbf{z}, \mathbf{u}, \mathbf{v})\|p_\theta(\mathbf{x}, \mathbf{y}, \mathbf{z}, \mathbf{u}, \mathbf{v})) + h(q(\mathbf{x}, \mathbf{y})) = \mathbb{E}_{q(\mathbf{x},\mathbf{y})}\Big[\mathcal{L}_{\theta,\phi}^{\mathrm{rec}}(\mathbf{X}, \mathbf{Y}) + \mathcal{L}_{\theta,\phi}^{\mathrm{reg}}(\mathbf{X}, \mathbf{Y})\Big],$$

where

$$\mathcal{L}_{\theta,\phi}^{\mathrm{rec}}(\mathbf{x}, \mathbf{y}) := \mathbb{E}_{q_\phi(\mathbf{z}|\mathbf{x},\mathbf{y})}\left[\int q_\phi(\mathbf{u}|\mathbf{Z}, \mathbf{x}) \log \frac{1}{p_\theta(\mathbf{x}|\mathbf{Z}, \mathbf{u})} \, \mathrm{d}\mathbf{u} + \int q_\phi(\mathbf{v}|\mathbf{Z}, \mathbf{y}) \log \frac{1}{p_\theta(\mathbf{y}|\mathbf{Z}, \mathbf{v})} \, \mathrm{d}\mathbf{v}\right],$$

$$\begin{aligned}\mathcal{L}_{\theta,\phi}^{\mathrm{reg}}(\mathbf{x}, \mathbf{y}) :=&\ D_{\mathsf{KL}}(q_\phi(\mathbf{z}|\mathbf{x}, \mathbf{y})q_\phi(\mathbf{u}|\mathbf{z}, \mathbf{x})q_\phi(\mathbf{v}|\mathbf{z}, \mathbf{y})\|p_\theta(\mathbf{z})p_\theta(\mathbf{u})p_\theta(\mathbf{v})) \\ =&\ D_{\mathsf{KL}}(q_\phi(\mathbf{z}|\mathbf{x}, \mathbf{y})\|p_\theta(\mathbf{z})) \\ &+ \mathbb{E}_{q_\phi(\mathbf{z}|\mathbf{x},\mathbf{y})}\Big[D_{\mathsf{KL}}(q_\phi(\mathbf{u}|\mathbf{Z}, \mathbf{x}))\|p_\theta(\mathbf{u})) + D_{\mathsf{KL}}(q_\phi(\mathbf{v}|\mathbf{Z}, \mathbf{y}))\|p_\theta(\mathbf{v}))\Big].\end{aligned}$$

We remark that a $\beta$-VAE (Higgins et al., 2017) type regularization in our joint model corresponds to imposing an additional weight $\beta > 1$ on $\mathcal{L}_{\theta,\phi}^{\mathrm{reg}}(\mathbf{x}, \mathbf{y})$ such that the objective function becomes $\mathbb{E}_{q(\mathbf{x},\mathbf{y})}[\mathcal{L}_{\theta,\phi}^{\mathrm{rec}}(\mathbf{X}, \mathbf{Y}) + \beta\mathcal{L}_{\theta,\phi}^{\mathrm{reg}}(\mathbf{X}, \mathbf{Y})]$. We note that

$$I_q(\mathbf{X}, \mathbf{Y}; \mathbf{Z}, \mathbf{U}, \mathbf{V}) \leq I_q(\mathbf{X}, \mathbf{Y}; \mathbf{Z}, \mathbf{U}, \mathbf{V}) + D_{\mathsf{KL}}(\tilde{q}_\phi(\mathbf{z}, \mathbf{u}, \mathbf{v})\|p_\theta(\mathbf{z})p_\theta(\mathbf{u})p_\theta(\mathbf{v})) \tag{37}$$

$$= \mathbb{E}_{q(\mathbf{x},\mathbf{y})}[D_{\mathsf{KL}}(q_\phi(\mathbf{z}|\mathbf{X}, \mathbf{Y})q_\phi(\mathbf{u}|\mathbf{z}, \mathbf{X})q_\phi(\mathbf{v}|\mathbf{z}, \mathbf{Y})\|p_\theta(\mathbf{z})p_\theta(\mathbf{u})p_\theta(\mathbf{v}))] \tag{38}$$

$$= \mathbb{E}_{q(\mathbf{x},\mathbf{y})}[\mathcal{L}_{\theta,\phi}^{\mathrm{reg}}(\mathbf{X}, \mathbf{Y})]. \tag{39}$$

Therefore, applying $\beta$-VAE type regularization imposes an additional regularization on $I_q(\mathbf{X}, \mathbf{Y}; \mathbf{Z}, \mathbf{U}, \mathbf{V})$, which corresponds to the entire latent bottleneck $(\mathbf{U}, \mathbf{V}, \mathbf{Z})$ for the information flow from $(\mathbf{X}, \mathbf{Y})$.

### B.3 ESTIMATION OF MUTUAL INFORMATION

With the typical Gaussian parameterization as presented above, we have an easy estimate for $I_q(\mathbf{X}, \mathbf{Y}; \mathbf{Z})$. After training, given a test dataset $\{\mathbf{x}_i, \mathbf{y}_i\}_{i=1}^N$, the mutual information $I_q(\mathbf{X}, \mathbf{Y}; \mathbf{Z})$ can be estimated as

$$
\begin{aligned}
I_q(\mathbf{X}, \mathbf{Y}; \mathbf{Z}) &= h_q(\mathbf{Z}) - h_q(\mathbf{Z}|\mathbf{X}, \mathbf{Y}) \\
&\approx h(p_\theta(\mathbf{z})) - \int q(\mathbf{x}, \mathbf{y}) h(q_\phi(\mathbf{z}|\mathbf{x}, \mathbf{y})) \, \mathrm{d}\mathbf{z} \\
&\approx h(p_\theta(\mathbf{z})) - \frac{1}{N} \sum_{i=1}^N h(q_\phi(\mathbf{z}|\mathbf{x}_i, \mathbf{y}_i)) \\
&= \frac{1}{2} \sum_{j=1}^{|\mathbf{Z}|} \log \boldsymbol{\sigma}_{0,\theta,j}^2 - \frac{1}{2N} \sum_{i=1}^N \sum_{j=1}^{|\mathbf{Z}|} \log \boldsymbol{\sigma}_{012,\phi,j}^2(\mathbf{x}_i, \mathbf{y}_i).
\end{aligned}
\tag{40}
$$

We can also estimate the variational upper bound of $I_q(\mathbf{X}, \mathbf{Y}; \mathbf{Z})$ in (11) as follows:

$$
\begin{aligned}
I_q(\mathbf{X}, \mathbf{Y}; \mathbf{Z}) \le \overline{I}_q(\mathbf{X}, \mathbf{Y}; \mathbf{Z}) &= \mathbb{E}_{q(\mathbf{x}, \mathbf{y})}[D_{\mathsf{KL}}(q_\phi(\mathbf{z}|\mathbf{X}, \mathbf{Y}) \| p_\theta(\mathbf{z}))] \\
&\approx \frac{1}{N} \sum_{i=1}^N D_{\mathsf{KL}}(q_\phi(\mathbf{z}|\mathbf{x}_i, \mathbf{y}_i) \| p_\theta(\mathbf{z})).
\end{aligned}
\tag{41}
$$

In the last expression, each KL divergence can be explicitly obtained from the Gaussian parameters of $q_\phi(\mathbf{z}|\mathbf{x}_i, \mathbf{y}_i)$ and $p_\theta(\mathbf{z})$.

## C  A DEEPER LOOK ON RELATED WORK

For the standard VAE (Kingma and Welling, 2014; Rezende et al., 2014), we refer the interested reader to Appendix A. In what follows, we revisit and decompose each VAE-type model into its encoder/prior/decoder components, and express the objective function in the form of the reverse KL divergence $D_{\mathsf{KL}}(q_\phi \| p_\theta)$, where $q_\phi$ is the joint distribution over the data variables and the latent variables defined by the data distribution and the encoders, and $p_\theta$ is defined by the priors and the decoders.

**JVAE/JMVAE**  JVAE (Vedantam et al., 2018) and JMVAE (Suzuki et al., 2016) consist of

- the joint encoder $q_\phi(\mathbf{w}|\mathbf{x}, \mathbf{y})$, the marginal encoders $q_\phi(\mathbf{w}|\mathbf{x}), q_\phi(\mathbf{w}|\mathbf{y})$,
- the prior $p_\theta(\mathbf{w})$,
- the decoders $p_\theta(\mathbf{x}|\mathbf{w}), p_\theta(\mathbf{y}|\mathbf{w})$.[1]

They share the same joint model objective

$$
D_{\mathsf{KL}}(q(\mathbf{x}, \mathbf{y}) q_\phi(\mathbf{w}|\mathbf{x}, \mathbf{y}) \| p_\theta(\mathbf{w}) p_\theta(\mathbf{x}|\mathbf{w}) p_\theta(\mathbf{y}|\mathbf{w})),
\tag{42}
$$

but differ in training the marginal encoders. JMVAE trains the marginal encoder $q_\phi(\mathbf{z}|\mathbf{x})$ via minimizing

$$
\mathbb{E}_{q(\mathbf{x}, \mathbf{y})}[D_{\mathsf{KL}}(q_\phi(\mathbf{w}|\mathbf{x}, \mathbf{y}) \| q_\phi(\mathbf{w}|\mathbf{x}))]
\tag{43}
$$

together with the joint model objective (42) (by adding two objective functions with an additional weight as a hyperparameter), while JVAE trains the marginal encoders separately from the joint model by training the *marginal VAE* as we proposed, i.e.,

$$
\underset{q_\phi(\mathbf{w}|\mathbf{x})}{\text{minimize}} \; D_{\mathsf{KL}}(q(\mathbf{x}) q_\phi(\mathbf{w}|\mathbf{x}) \| p_\theta(\mathbf{w}) p_\theta(\mathbf{x}|\mathbf{w})).
\tag{44}
$$

---

[1] We remark that $p_\theta(\mathbf{x}|\mathbf{w})$ and $p_\theta(\mathbf{y}|\mathbf{w})$ are fully characterized by the deterministic decoders $\mathbf{x}_\theta(\mathbf{w})$ and $\mathbf{y}_\theta(\mathbf{w})$.

It is worthwhile to compare the regularization term $I_q(\mathbf{X}, \mathbf{Y}; \mathbf{Z})$ in Wyner VAE with the idea of $\beta$-VAE (Higgins et al., 2017), which empirically showed that an additional weight on the regularization term finds a disentangled representation in a VAE model. If we apply a $\beta$-VAE type regularization in our joint model, then it corresponds to an additional weight $\beta > 1$ on the mutual information on the joint representation ($I_q(\mathbf{X}, \mathbf{Y}; \mathbf{Z}, \mathbf{U}, \mathbf{V})$), not only on the common representation ($I_q(\mathbf{X}, \mathbf{Y}; \mathbf{Z})$) as in Wyner VAE. (See Appendix B.2.) In words, Wyner VAE provides a finer control on the information flow from $(\mathbf{X}, \mathbf{Y})$, by manipulating the common path $q_\phi(\mathbf{z}|\mathbf{x}, \mathbf{y})$, while $\beta$-VAE blocks the entire latent bottleneck $q_\phi(\mathbf{z}|\mathbf{x}, \mathbf{y})q_\phi(\mathbf{u}|\mathbf{z}, \mathbf{x})q_\phi(\mathbf{v}|\mathbf{z}, \mathbf{y})$.

**CVAE**    CVAE (Sohn et al., 2015) for modeling $q(\mathbf{y}|\mathbf{x})$ consists of

- the encoder $q_\phi(\mathbf{v}|\mathbf{y}, \mathbf{x})$,
- the prior $p_\theta(\mathbf{v}|\mathbf{x})$,
- the decoder $p_\theta(\mathbf{y}|\mathbf{v}, \mathbf{x})$.

The objective function is then given as
$$\mathbb{E}_{q(\mathbf{x})}[D_{\mathsf{KL}}(q(\mathbf{y}|\mathbf{x})q_\phi(\mathbf{v}|\mathbf{y}, \mathbf{x})\|p_\theta(\mathbf{v}|\mathbf{x})p_\theta(\mathbf{y}|\mathbf{v}, \mathbf{x}))]. \tag{45}$$
Note that without the conditioning variable $\mathbf{x}$, this model boils down to the vanilla VAE for modeling $q(\mathbf{y})$. If we assume a Markov chain $\mathbf{X} - \mathbf{V} - \mathbf{Y}$, the decoder $p_\theta(\mathbf{y}|\mathbf{v}, \mathbf{x})$ can be replaced with $p_\theta(\mathbf{y}|\mathbf{v})$.

We performed conditional generation with style control with CVAE as follows: Given a style reference $\mathbf{y}_0$ and its corresponding $\mathbf{x}_0$, we sample and keep $\mathbf{v}_0$ via the encoder $q_\phi(\mathbf{v}|\mathbf{y}, \mathbf{x})$. Given a new $\mathbf{x}_1$, we take $\mathbf{y}_{0,1} = \mathbf{y}_\theta(\mathbf{v}_0, \mathbf{x}_1)$ as a new sample. Note that this scheme assumes that $\mathbf{v}_0 \sim q_\phi(\mathbf{v}|\mathbf{y}_0, \mathbf{x}_0)$ captures only the remaining information in $\mathbf{y}_0$ excluding the information on $\mathbf{x}_0$.

**SSVAE/ADGM**    Semi-supervised VAE (SSVAE) (Kingma et al., 2014) proposed a similar conditional VAE model for modeling $q(\mathbf{y}|\mathrm{x})$ especially when the conditioning variable $\mathrm{x}$ is discrete. SSVAE consists of

- the encoders $q_\phi(\mathbf{v}|\mathbf{y}, \mathrm{x})$, $q_\phi(\mathrm{x}|\mathbf{y})$ (classifier),
- the priors $p_\theta(\mathbf{v})$, $p_\theta(\mathrm{x})$,
- the decoder $p_\theta(\mathbf{y}|\mathbf{v}, \mathrm{x})$.

Note that it has an additional encoder $q_\phi(\mathrm{x}|\mathbf{y})$ (classifier) on top of $q_\phi(\mathbf{v}|\mathbf{y}, \mathrm{x})$ from CVAE, and assume the priors $p_\theta(\mathbf{v})$ and $p_\theta(\mathrm{x})$, which replace the conditional prior $p_\theta(\mathbf{v}|\mathrm{x})$ and the data distribution $q(\mathrm{x})$ in CVAE, respectively. With a paired (i.e., labelled) data, SSVAE minimizes
$$D_{\mathsf{KL}}(q(\mathrm{x}, \mathbf{y})q_\phi(\mathbf{v}|\mathbf{y}, \mathrm{x})\|p_\theta(\mathrm{x})p_\theta(\mathbf{v})p_\theta(\mathbf{y}|\mathbf{v}, \mathrm{x})). \tag{46}$$
With an unlabeled data, SSVAE minimizes
$$D_{\mathsf{KL}}(q(\mathbf{y})q_\phi(\mathrm{x}|\mathbf{y})q_\phi(\mathbf{v}|\mathbf{y}, \mathrm{x})\|p_\theta(\mathrm{x})p_\theta(\mathbf{v})p_\theta(\mathbf{y}|\mathbf{v}, \mathrm{x})). \tag{47}$$
Note that the label information $q(\mathrm{x})$ in (46) is replaced with $q_\phi(\mathrm{x}|\mathbf{y})$ in (47) as the label information is missing in the unlabeled data.

Formally, in this degenerate case, identifying the conditioning variable $\mathbf{X}$ as the common latent variable $\mathbf{Z}$ in Wyner VAE recovers SSVAE. Yet, SSVAE was proposed in the context of (semi-supervised) classification problem which aims to learn a classifier $q_\phi(\mathrm{x}|\mathbf{y})$, and for a high-dimensional $\mathbf{X}$, it is not feasible to directly model the conditional distribution $q_\phi(\mathbf{x}|\mathbf{y})$.

Auxiliary deep generative model (ADGM) (Maaløe et al., 2016) adds an auxiliary latent variable $\mathbf{Z}$ to SSVAE to improve the performance. ADGM consists of

- the encoders $q_\phi(\mathbf{z}|\mathbf{y})$, $q_\phi(\mathbf{v}|\mathbf{z}, \mathbf{y}, \mathrm{x})$, $q_\phi(\mathrm{x}|\mathbf{z}, \mathbf{y})$ (classifier),
- the priors $p_\theta(\mathbf{v})$, $p_\theta(\mathrm{x})$,
- the decoders $p_\theta(\mathbf{y}|\mathbf{v}, \mathrm{x})$, $p_\theta(\mathbf{z}|\mathbf{v}, \mathrm{x}, \mathbf{y})$.

Note the new components $q_\phi(\mathbf{z}|\mathbf{y})$ and $p_\theta(\mathbf{z}|\mathbf{v}, \mathrm{x}, \mathbf{y})$ on top of the SSVAE, and the original encoder/decoder components have additional condition on the auxiliary variable $\mathbf{Z}$. However, as ADGM does not impose any additional conditional independence with $\mathbf{Z}$, it is not directly comparable to Wyner VAE.

**VCCA-private**    As noted earlier, VCCA-private (Wang et al., 2016) consists of

- the encoders $q_\phi(\mathbf{z}|\mathbf{x})$, (and/or $q_\phi(\mathbf{z}|\mathbf{y})$), $q_\phi(\mathbf{u}|\mathbf{x})$, $q_\phi(\mathbf{v}|\mathbf{y})$,
- the priors $p_\theta(\mathbf{z})$, $p_\theta(\mathbf{u})$, $p_\theta(\mathbf{v})$,
- the decoders $p_\theta(\mathbf{x}|\mathbf{z}, \mathbf{u})$, $p_\theta(\mathbf{y}|\mathbf{z}, \mathbf{v})$.

Note that the prior and the decoder components are same with Wyner VAE. Hence, the objective for VCCA-private for the marginal encoder $q_\phi(\mathbf{z}|\mathbf{x})$ can be expressed as

$$D_{\mathsf{KL}}(q_\phi(\mathbf{z}|\mathbf{x})q_\phi(\mathbf{u}|\mathbf{x})q_\phi(\mathbf{v}|\mathbf{y})\|p_\theta(\mathbf{u})p_\theta(\mathbf{v})p_\theta(\mathbf{z})p_\theta(\mathbf{x}|\mathbf{z}, \mathbf{u})p_\theta(\mathbf{y}|\mathbf{z}, \mathbf{v})). \tag{48}$$

To model the other direction of the marginal encoder $q_\phi(\mathbf{y}|\mathbf{z})$, they minimize

$$D_{\mathsf{KL}}(q_\phi(\mathbf{z}|\mathbf{y})q_\phi(\mathbf{u}|\mathbf{x})q_\phi(\mathbf{v}|\mathbf{y})\|p_\theta(\mathbf{u})p_\theta(\mathbf{v})p_\theta(\mathbf{z})p_\theta(\mathbf{x}|\mathbf{z}, \mathbf{u})p_\theta(\mathbf{y}|\mathbf{z}, \mathbf{v})). \tag{49}$$

To learn $q_\phi(\mathbf{z}|\mathbf{x})$, $q_\phi(\mathbf{z}|\mathbf{y})$ simultaneously, BiVCCA-private minimizes a convex combination of the two KL divergence terms.

We performed conditional generation with style control with VCCA-private as follows: Given a style reference $\mathbf{y}_0$, we sample and keep $\mathbf{v}_0$ via the encoder $q_\phi(\mathbf{v}|\mathbf{y})$. Given a new attribute $\mathbf{x}_1$, we take $\mathbf{y}_{0,1} = \mathbf{y}_\theta(\mathbf{v}_0, \mathbf{x}_1)$ as a new sample.

**VIB**    VIB (Alemi et al., 2017) proposed a variational relaxation of the following minimization problem posed by the information bottleneck principle (Tishby et al., 1999):

$$\underset{q_\phi(\mathbf{z}|\mathbf{x})}{\text{minimize}} \ \beta I_q(\mathbf{X}; \mathbf{Z}) - I_q(\mathbf{Y}; \mathbf{Z}), \tag{50}$$

where $(\mathbf{X}, \mathbf{Y}, \mathbf{Z}) \sim q(\mathbf{x}, \mathbf{y})q_\phi(\mathbf{z}|\mathbf{x})$. VIB introduces two variational distributions $p_\theta(\mathbf{z})$ and $p_\theta(\mathbf{y}|\mathbf{z})$ that approximate $\tilde{q}_\phi(\mathbf{z})$ and $\tilde{q}_\phi(\mathbf{y}|\mathbf{z})$. Then, based on standard variational bounds on the mutual information terms (see, e.g., Zhao et al. (2018)), we obtain a variational upper bound on $\beta I_q(\mathbf{X}; \mathbf{Z}) - I_q(\mathbf{Y}; \mathbf{Z})$ as

$$\beta \mathbb{E}_{q(\mathbf{x})}[D(q_\phi(\mathbf{z}|\mathbf{X})\|p_\theta(\mathbf{z}))] - \mathbb{E}_{q(\mathbf{x},\mathbf{y})q_\phi(\mathbf{z}|\mathbf{x})}\left[\log \frac{1}{p_\theta(\mathbf{Y}|\mathbf{Z})}\right] + h(q(\mathbf{y})), \tag{51}$$

which is the objective function for VIB. Note that the relaxation gap is given by

$$\beta D(\tilde{q}_\phi(\mathbf{z})\|p_\theta(\mathbf{z})) + \mathbb{E}_{\tilde{q}_\phi(\mathbf{z})}[D(\tilde{q}_\phi(\mathbf{y}|\mathbf{Z})\|p_\theta(\mathbf{y}|\mathbf{Z}))]. \tag{52}$$

# D    LIKELIHOOD ESTIMATION

For each likelihood of our interest, we derive a naive Monte Carlo (MC) estimator and an MC estimator with importance sampling (see, e.g., Rubinstein and Kroese (2016)). Here we only present the estimators for Wyner VAE, but the estimators for other models can be derived in the same manner.

## D.1    JOINT LIKELIHOOD

We wish to estimate the joint log-likelihood of the model with respect to given test data $\{(\mathbf{x}^{(i)}, \mathbf{y}^{(i)})\}_{i=1}^N$, i.e., $\sum_{i=1}^N \log p_\theta(\mathbf{x}^{(i)}, \mathbf{y}^{(i)})$, where

$$p_\theta(\mathbf{x}, \mathbf{y}) = \int p_\theta(\mathbf{z})p_\theta(\mathbf{u})p_\theta(\mathbf{v})p_\theta(\mathbf{x}|\mathbf{z}, \mathbf{u})p_\theta(\mathbf{y}|\mathbf{z}, \mathbf{v}) \, \mathrm{d}\mathbf{z} \, \mathrm{d}\mathbf{u} \, \mathrm{d}\mathbf{v}. \tag{53}$$

(1) Monte Carlo estimator: Let $(\mathbf{z}^{(s)}, \mathbf{u}^{(s)}, \mathbf{v}^{(s)}) \sim p_\theta(\mathbf{z})p_\theta(\mathbf{u})p_\theta(\mathbf{v})$ for $s = 1, \ldots, S$.

$$\hat{p}_\theta(\mathbf{x}, \mathbf{y}) = \frac{1}{S} \sum_{s=1}^S p_\theta(\mathbf{x}|\mathbf{z}^{(s)}, \mathbf{u}^{(s)})p_\theta(\mathbf{y}|\mathbf{z}^{(s)}, \mathbf{v}^{(s)}). \tag{54}$$

(2) Importance sampling: For each $(\mathbf{x}, \mathbf{y})$, let $(\mathbf{z}^{(s)}, \mathbf{u}^{(s)}, \mathbf{v}^{(s)}) \sim q_\phi(\mathbf{z}|\mathbf{x}, \mathbf{y})q_\phi(\mathbf{u}|\mathbf{z}, \mathbf{x})q_\phi(\mathbf{v}|\mathbf{z}, \mathbf{y})$ for $s = 1, \ldots, S$.

$$\hat{p}_\theta(\mathbf{x}, \mathbf{y}) = \frac{1}{S} \sum_{s=1}^S \frac{p_\theta(\mathbf{z}^{(s)})p_\theta(\mathbf{u}^{(s)})p_\theta(\mathbf{v}^{(s)})p_\theta(\mathbf{x}|\mathbf{z}^{(s)}, \mathbf{u}^{(s)})p_\theta(\mathbf{y}|\mathbf{z}^{(s)}, \mathbf{v}^{(s)})}{q_\phi(\mathbf{z}^{(s)}|\mathbf{x}, \mathbf{y})q_\phi(\mathbf{u}^{(s)}|\mathbf{z}^{(s)}, \mathbf{x})q_\phi(\mathbf{v}^{(s)}|\mathbf{z}^{(s)}, \mathbf{y})}. \tag{55}$$

### D.2 CONDITIONAL LIKELIHOOD

We wish to estimate the conditional log-likelihood of the conditional path of Wyner VAE from $\mathbf{x}$ to $\mathbf{y}$, i.e., $q_\phi(\mathbf{z}|\mathbf{x})p_\theta(\mathbf{v})p_\theta(\mathbf{y}|\mathbf{z},\mathbf{v})$, with respect to given test data $\{(\mathbf{x}^{(i)},\mathbf{y}^{(i)})\}_{i=1}^N$, i.e.,

$$\sum_{i=1}^N \log r_{\theta,\phi}(\mathbf{y}^{(i)}|\mathbf{x}^{(i)}), \tag{56}$$

where

$$r_{\theta,\phi}(\mathbf{y}|\mathbf{x}) = \int q_\phi(\mathbf{z}|\mathbf{x})p_\theta(\mathbf{v})p_\theta(\mathbf{y}|\mathbf{z},\mathbf{v})\,\mathrm{d}\mathbf{v}\,\mathrm{d}\mathbf{z}. \tag{57}$$

(1) Monte Carlo estimation: Given $\mathbf{x}$, let $(\mathbf{z}^{(s)},\mathbf{v}^{(s)}) \sim q_\phi(\mathbf{z}|\mathbf{x})p_\theta(\mathbf{v})$ for $s = 1,\ldots,S$.

$$\hat{r}_{\theta,\phi}(\mathbf{y}|\mathbf{x}) = \frac{1}{S}\sum_{s=1}^S p_\theta(\mathbf{y}|\mathbf{z}^{(s)},\mathbf{v}^{(s)}). \tag{58}$$

(2) Importance sampling: For each $(\mathbf{x},\mathbf{y})$, let $(\mathbf{z}^{(s)},\mathbf{v}^{(s)}) \sim q_\phi(\mathbf{z}|\mathbf{x},\mathbf{y})q_\phi(\mathbf{v}|\mathbf{z},\mathbf{y})$ for $s = 1,\ldots,S$.

$$\hat{r}_{\theta,\phi}(\mathbf{y}|\mathbf{x}) = \frac{1}{S}\sum_{s=1}^S \frac{q_\phi(\mathbf{z}^{(s)}|\mathbf{x})p_\theta(\mathbf{v}^{(s)})p_\theta(\mathbf{y}|\mathbf{z}^{(s)},\mathbf{v}^{(s)})}{q_\phi(\mathbf{z}^{(s)}|\mathbf{x},\mathbf{y})q_\phi(\mathbf{v}^{(s)}|\mathbf{z}^{(s)},\mathbf{y})}. \tag{59}$$

## E ADDITIONAL EXPERIMENTAL RESULTS

### E.1 MIXTURE OF GAUSSIANS

Here we present the test log-likelihood evaluation of JVAE and VIB in conditional generation tasks of the mixture of Gaussians (MoG) dataset; recall Fig. 4. Both JVAE and VIB performed worse as the training epochs increased, that is, they overfit to the training data (Fig. 8). We visually illustrate how they failed in Fig. 9. Fig. 9(a) shows the outlook of the MoG dataset in our experiment. Fig. 9 (e1,e2) show that JVAE captured all the components at the beginning, but then collapsed to a few components afterwards. On the other hand, Fig. 9 (f1,f2) show that VIB only captured the average behaviors, although gradually adapting to the underlying data.

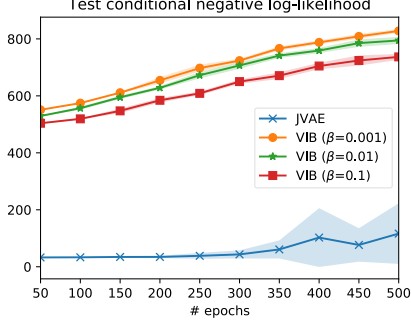

Figure 8: Conditional nll values of JVAE and VIB for MoG dataset. For each point of the plots, we trained 10 different models and plotted average values with the shaded region that shows the standard deviation. (Two largest and smallest outliers were dropped for each point.)

---

[2]All the scatter plots were generated based on the Gaussian kernel density estimation.

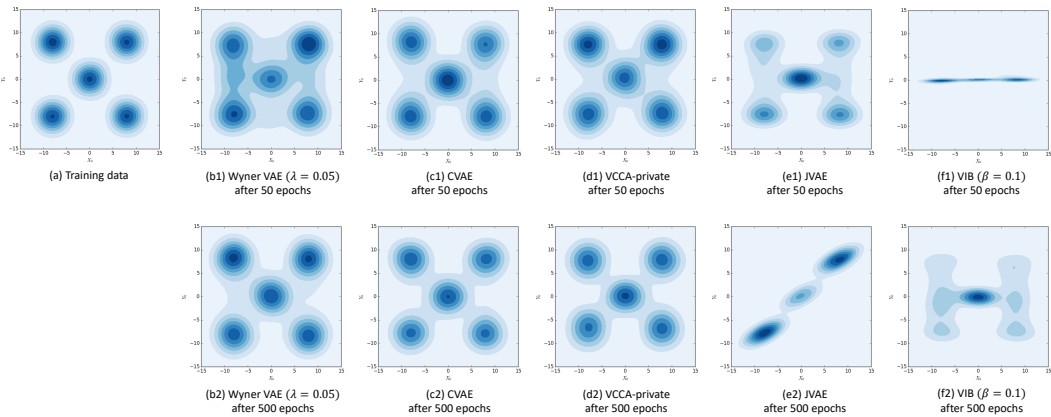

Figure 9: Visualization of conditionally generated samples for MoG dataset. Each axis of the scatter plots corresponds to the first coordinate of $\mathbf{X}_i$ and $\mathbf{Y}_i$, respectively.[2] The $\mathbf{X}$ data points were from the test data, and the $\mathbf{Y}$ data points were generated from the conditional models based on the test data. One sample was generated for each data point.

## E.2   MNIST–MNIST ADD-1

Table 5: Accompanying table for Fig. 5: Summary of numerical evaluations of MNIST–MNIST add-1 experiments. For each row, we trained 10 different models and dropped two outliers for each average and standard deviation.

| | Joint nll | Conditional nll | $I_q(\mathbf{X}, \mathbf{Y}; \mathbf{Z})$ | Accuracy (%) | Variance |
|---|---|---|---|---|---|
| JVAE (Vedantam et al., 2018) | $\mathbf{1173.40 \pm 13.98}$ | $514.26 \pm 12.03$ | $65.89 \pm 0.51$ | $\mathbf{98.84 \pm 0.11}$ | $0.0032 \pm 0.0005$ |
| Wyner VAE ($\lambda = 0$) | $1198.95 \pm 5.20$ | $172.90 \pm 45.18$ | $41.35 \pm 0.78$ | $98.01 \pm 0.15$ | $0.0254 \pm 0.0036$ |
| Wyner VAE ($\lambda = 0.05$) | $1195.09 \pm 10.35$ | $58.86 \pm 21.78$ | $33.57 \pm 0.84$ | $97.46 \pm 0.86$ | $0.0353 \pm 0.0027$ |
| Wyner VAE ($\lambda = 0.10$) | $1212.66 \pm 10.61$ | $\mathbf{-0.51 \pm 6.94}$ | $24.52 \pm 0.89$ | $91.33 \pm 1.31$ | $0.0454 \pm 0.0009$ |
| Wyner VAE ($\lambda = 0.15$) | $1220.64 \pm 10.71$ | $\mathbf{3.35 \pm 6.04}$ | $19.73 \pm 0.78$ | $84.90 \pm 2.68$ | $0.0480 \pm 0.0006$ |
| Wyner VAE ($\lambda = 0.20$) | $1230.32 \pm 9.30$ | $16.37 \pm 7.06$ | $16.04 \pm 0.50$ | $79.94 \pm 2.83$ | $0.0494 \pm 0.0006$ |
| VCCA-private (Wang et al., 2016) | $1254.89 \pm 7.17$ | $90.64 \pm 5.50$ | - | $59.63 \pm 1.25$ | $\mathbf{0.0548 \pm 0.0003}$ |
| CVAE (Sohn et al., 2015) | - | $15.39 \pm 6.41$ | - | $97.69 \pm 0.28$ | $0.0404 \pm 0.0004$ |
| VIB (Alemi et al., 2017) ($\beta = 0.001$) | - | $733.86 \pm 13.88$ | - | $96.54 \pm 0.24$ | $0.0000 \pm 0.0000$ |

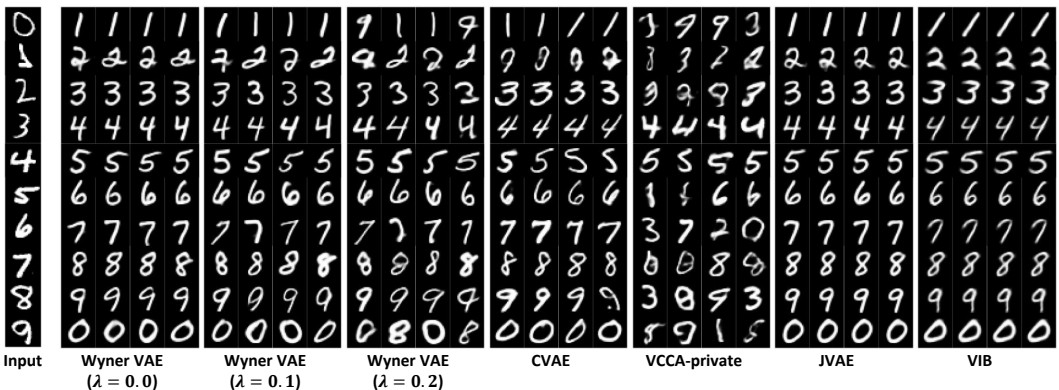

Figure 10: Conditional generation.

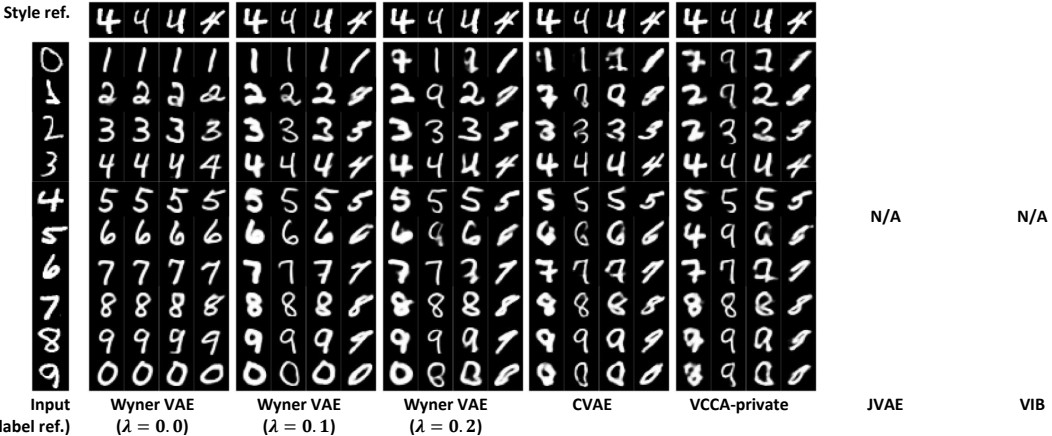

Figure 11: Conditional generation with style control.

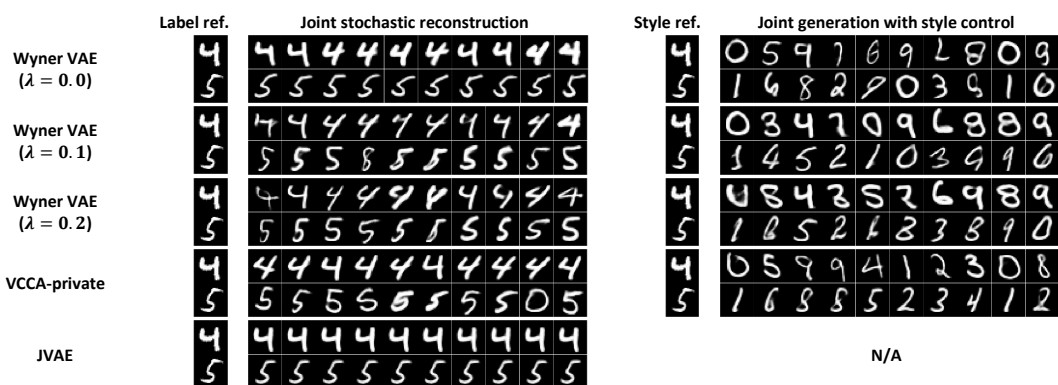

Figure 12: Joint generation.

### E.3 MNIST–SVHN

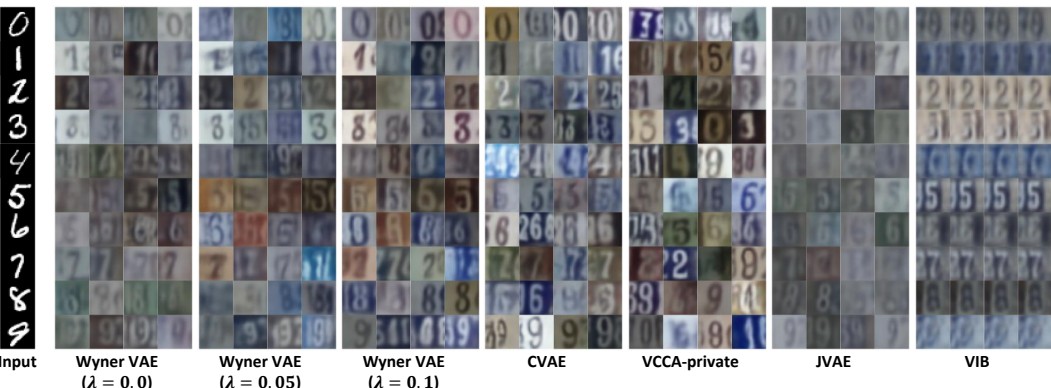

Figure 13: Conditional generation.

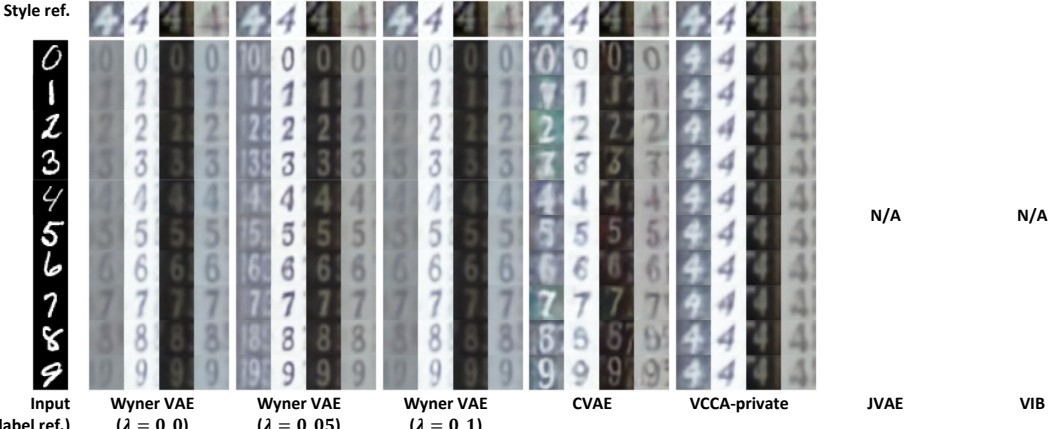

Figure 14: Conditional generation with style control.

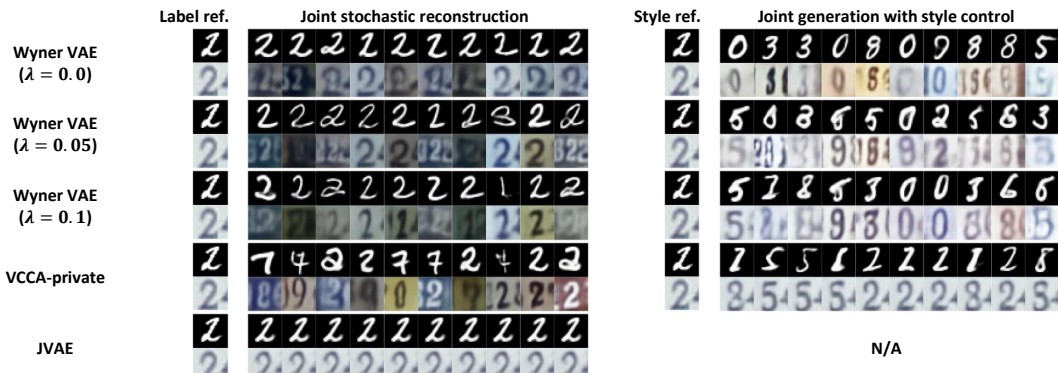

Figure 15: Joint generation.

### E.4 (FACE, ATTRIBUTE) PAIRS FROM CELEBA DATASET

CelebA dataset (Liu et al., 2015) consists of pairs of a face image and a 40-dim. binary vector that contains attributes information of the face. We performed conditional generation of face images $(\mathbf{Y})$ given an attribute vector $(\mathbf{X})$. Since an attribute $\mathbf{X}$ is a function of a given face image $\mathbf{Y}$, we let the dimension of the local variable $\mathbf{U}$ be 0 in this case. Fig. 16 presents samples of CelebA faces from Wyner VAE, JVAE, and CVAE — Wyner VAE with $\lambda = 0.1$ generated a variety of faces with the correct attributes, while JVAE generated images with little variations as previously observed and CVAE generated diverse images but often with wrong attributes. See also Appendix E.4.2 for the results with style control and a numerical evaluation of the performance of Wyner VAE on CelebA dataset.

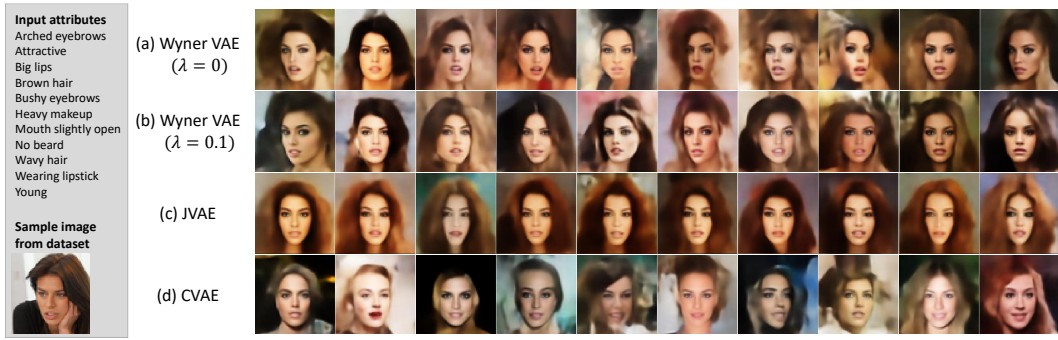

Figure 16: Samples from Wyner VAE, JVAE, and CVAE for CelebA dataset. Multiple face image samples were conditionally sampled given an attribute vector listed at the leftmost column.

Note that this is a special case where the target variable $\mathbf{Y}$ is a function of a conditioning variable $\mathbf{X}$, which is a degenerate case in the sense that any random variable $\mathbf{Z}$ that is bijective with $\mathbf{X}$ can serve as an optimal common representation that achieves $J(\mathbf{X}; \mathbf{Y})$, and Wyner VAE and CVAE may have comparable conditional generation performance. In practice, however, Wyner VAE outperforms CVAE since Wyner VAE learns a good representation $\mathbf{Z}$ of $\mathbf{X}$ that is helpful in generating $\mathbf{Y}$ conditionally, while CVAE directly uses the raw $\mathbf{X}$ for conditioning.

For a qualitative evidence, we present a few Attribute→Face samples in Fig. 17 from models trained with more data as in the paper, from a truly unseen attribute (likely-female-features + bald)—Wyner VAE can produce plausible images as it finds a good representation $\mathbf{Z}$ of an unseen attribute, while CVAE fails.

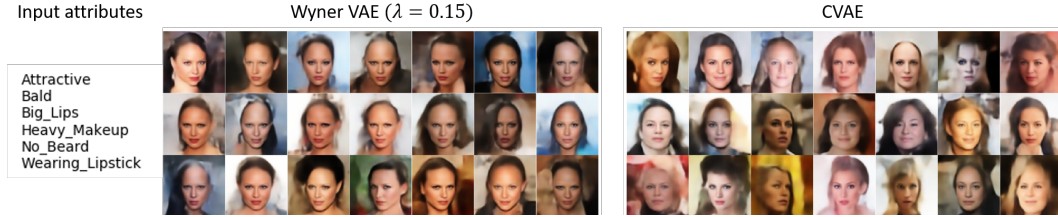

Figure 17: Attribute→Face (CelebA) samples from Wyner VAE and CVAE.

### E.4.1 NUMERICAL EVALUATION

We present an additional numerical evaluation of Wyner VAE to corroborate the effect of $\lambda > 0$. With CelebA models, we evaluated both conditional paths, i.e., face to attribute classification (face2attribute) and attribute to face generation (attribute2face). For the attribute classification, we counted the number of corrected classified binary attributes out of 40. The image variance is evaluated with per pixel, while all pixel values were normalized between 0 and 1. Table 6 summarizes the results. We can observe that both the accuracy and the variance for conditional generation from a given attribute were maximized the around $\lambda \sim 0.2$. Note that JVAE performs worse in classification

Table 6: Numerical evaluation of Wyner VAE and JVAE for CelebA dataset.

|  | Face2attribute accuracy (%) | Attribute2face variance |
|---|---|---|
| Wyner VAE ($\lambda = 0$) | 89.26 | 0.0482 |
| Wyner VAE ($\lambda = 0.05$) | 89.21 | 0.0493 |
| Wyner VAE ($\lambda = 0.1$) | 89.29 | 0.0516 |
| Wyner VAE ($\lambda = 0.15$) | **89.32** | 0.0527 |
| Wyner VAE ($\lambda = 0.2$) | 89.28 | **0.0544** |
| JVAE (Vedantam et al., 2018) | 88.11 | 0.0073 |

accuracy with a comparably very small per-pixel variance implying much less variations in the generated samples.

### E.4.2 ADDITIONAL ATTRIBUTE2FACE GENERATION RESULTS

Here we present additional conditional generated samples (with and without style control) from CelebA models. We used the sample images and attributes shown in Fig. 18 for these experiments.

For conditional generation in Fig. 19, samples were generated only based on the sample attributes.

For conditional generation with style control in Fig. 20, Wyner VAE and CVAE first extracted style information from sample images in the leftmost column, and then generated new samples from the original attribute added with a new binary attribute specified in the topmost row for each column. Hence, the second column corresponds to the reconstruction of the style reference images from the faces and the corresponding attributes. We remark that JVAE is not capable of style manipulation, and the results from conditional generation with JVAE are given as a reference.

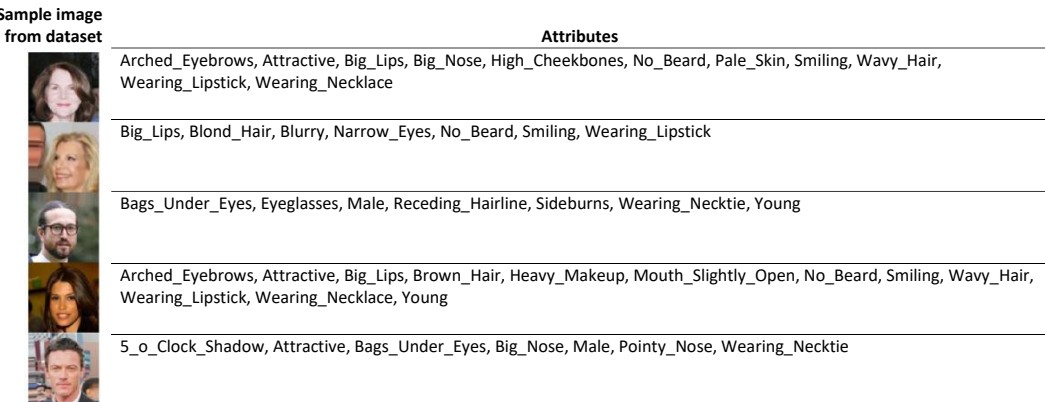

Figure 18: Sample images and their attribute vectors from CelebA dataset.

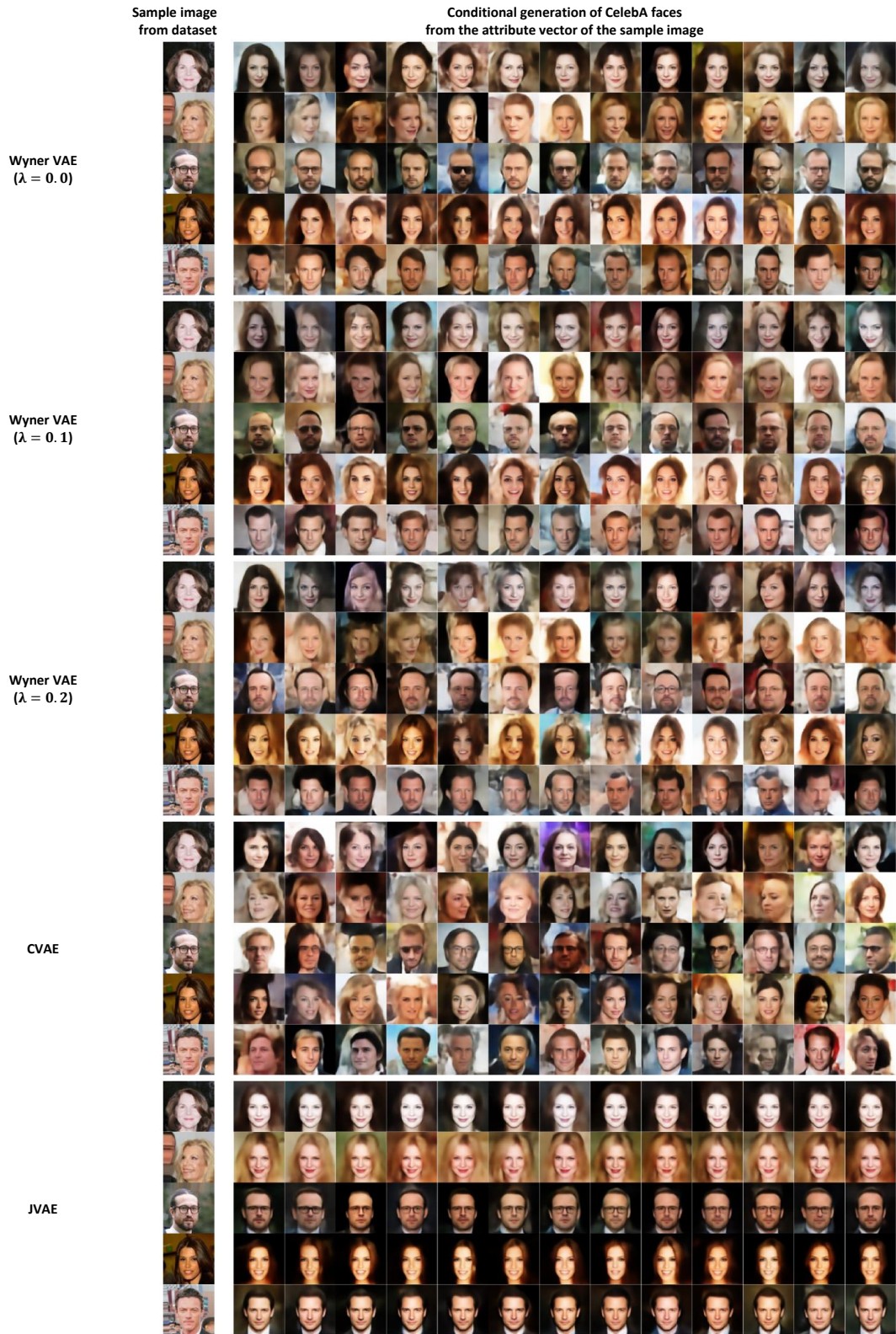

Figure 19: Conditional generation (attribute2face).

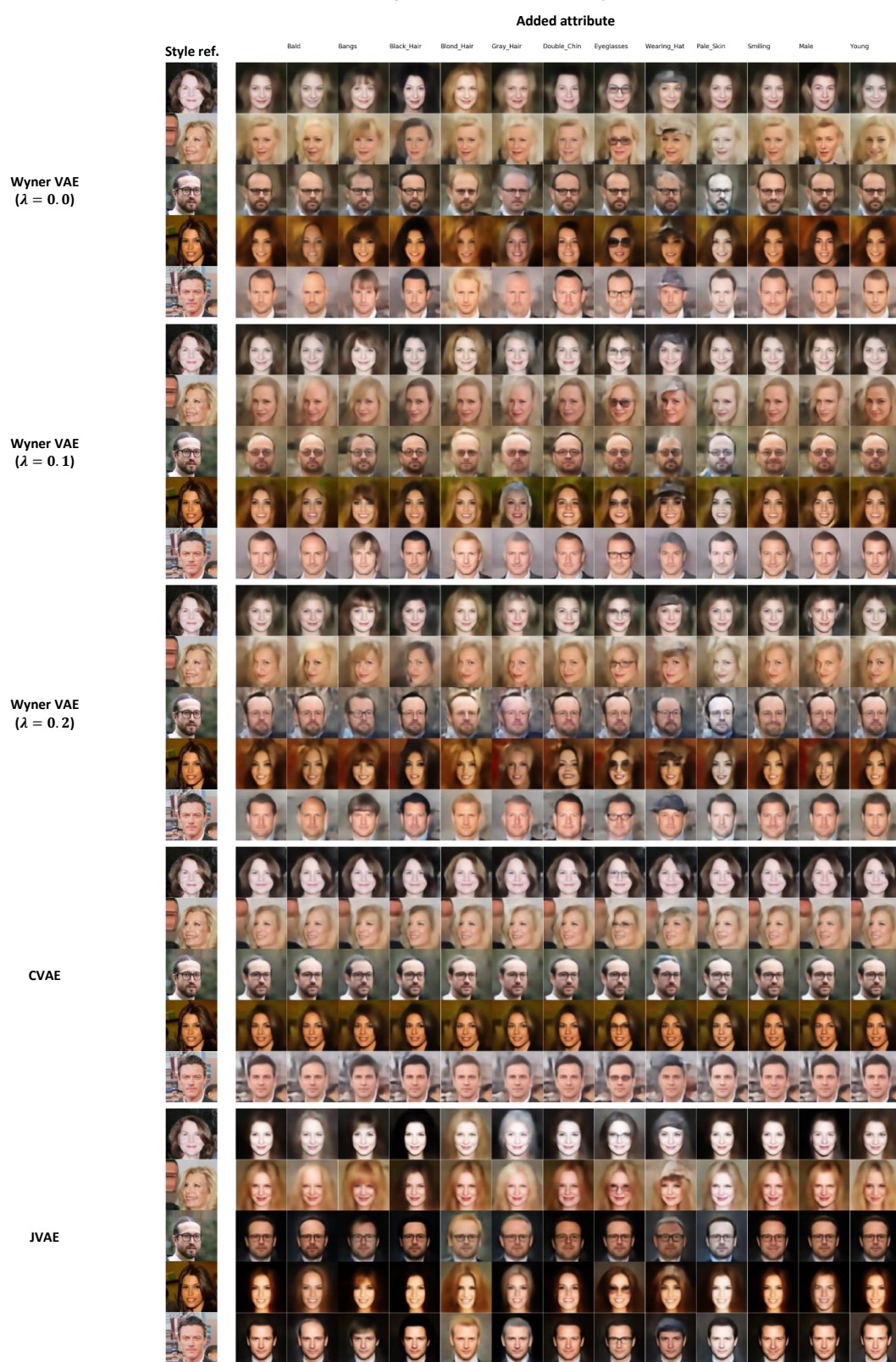

Figure 20: Conditional generation with style control (attribute2face). Note that JVAE is not capable of style manipulation, and the results were simply generated from attribute2face generation and are given as a reference. Hence, the leftmost column is the sample image as in the conditional generation experiment for JVAE.

## F    EXPERIMENT DETAILS

We used the same parameterization, same latent dimensions, and the same network architecture across the different models to be a fair comparison. For simplicity, we used the standard Gaussian parameterization of each component for all the implemented models as in the standrad VAE. (See Appendix A and Appendx B.) For the prior distributions $p_\theta(\mathbf{z}), p_\theta(\mathbf{u}), p_\theta(\mathbf{v})$, we let the isotropic variances $\boldsymbol{\sigma}^2_{0,\theta}, \boldsymbol{\sigma}^2_{1,\theta}, \boldsymbol{\sigma}^2_{2,\theta}$ be trainable. With this degree of freedom, the neural networks select necessary dimensions in the latent spaces over $\mathbf{Z}, \mathbf{U}, \mathbf{V}$ by assigning small variances to unused dimensions. For the MoG experiment, we used a constant 1/2 for the decoder variance, so that the log-loss corresponds to the $l_2$-squared loss. For the rest of the experiments, we allowed the diagonal variances to be trainable: we allocated one trainable decoder variance per channel, independent from the latent inputs. We found that this trick results in sharper images across the models.

We set the dimension of the latent variable $\mathbf{W}$ in JVAE as the sum of the dimensions of $\mathbf{Z}, \mathbf{U}, \mathbf{V}$ as $\mathbf{W}$ corresponds the joint representation. Similarly, since the latent variable of CVAE corresponds to the local randomness in Wyner VAE, we let the dimension of $\mathbf{V}$ in CVAE be equal to the dimension of $\mathbf{V}$ in Wyner VAE. We used (10,10,10), (32,32,32), (128,128,128), and (128,0,128) as the latent dimensions of $(\mathbf{Z}, \mathbf{U}, \mathbf{V})$ for MoG, MNIST–MNIST add-1, MNIST-SVHN, and CelebA, respectively.

All log-likelihood values in the experiments were estimated by importance sampling; see Appendix D. We used $S = 100$ importance samples for each data point. The mutual information $I_q(\mathbf{X}, \mathbf{Y}; \mathbf{Z})$ was estimated in a straightforward manner with the test dataset under the Gaussian parameterization; see Appendix B.

**Computing infrastructure**    We used NVIDIA TITAN X (Pascal) for our experiments.

**Implementation**    We implemented all models using Keras[3] with tensorflow backend.

**Datasets**    For the mixture of Gaussians dataset, we generate the paired dataset $(\mathbf{X}_i, \mathbf{Y}_i) \in \mathbb{R}^{10} \times \mathbb{R}^{10}$ as follows. Let $\mu \colon \{1, 2, 3, 4, 5\} \to \mathbb{R} \times \mathbb{R}$ be a function defined as

$$\mu(1) = (0,0), \mu(2) = (4,4), \mu(3) = (-4,4), \mu(4) = (-4,-4), \mu(5) = (4,-4).$$

Then, we let

$$\mathbf{X}_i = 2 \begin{bmatrix} \mu_1(Z_i)\mathbb{1}_5 + \mathbf{U}_i \\ \mu_1(Z_i)\mathbb{1}_5 - \mathbf{U}_i \end{bmatrix}, \quad \mathbf{Y}_i = 2 \begin{bmatrix} \mu_2(Z_i)\mathbb{1}_5 + \mathbf{V}_i \\ \mu_2(Z_i)\mathbb{1}_5 - \mathbf{V}_i \end{bmatrix}, \tag{60}$$

where $Z_i \sim \mathrm{Unif}(\{1,2,3,4,5\})$, $\mathbf{U}_i, \mathbf{V}_i \sim \mathcal{N}(0, I_5)$ are drawn independently. Here, $\mathbb{1}_5 \in \mathbb{R}^5$ denotes the all-1 vector.

For MNIST–MNIST add-1, we constructed 50k add-1 pairs from the MNIST training dataset. For MNIST–SVHN domain adapation, we constructed 50k MNIST-SVHN pairs from MNIST and SVHN training datasets. For testing, we similarly constructed 1k paired images from MNIST and SVHN test datasets in each case. For CelebA experiments, we set aside 5k samples for test dataset, and used the rest in training.

**Network architectures**    Let `c5s1-`$k$`-{activation}` denote a $5 \times 5$ Convolution–BatchNorm–activation with $k$ filters and stride $1 \times 1$. Let `d3s2-`$k$`-{activation}` / `u3s2-`$k$`-{activation}` denote $3 \times 3$ Convolution / Deconvolution–BatchNorm–activation with $k$ filters and stride $2 \times 2$, respectively. Let `res-`$k$ be a residual block that contains two $3 \times 3$ convolutional layers with $k$ filters in each (i.e., `c3s1-`$k$`-LReLU, c3s1-`$k$) and a skip connection from the input to the output. Let `fc-`$k$`-{activation}` be a fully-connected layer with $k$ units and a non-linear activation.

For MNIST–MNIST add-1, MNIST-SVHN, and CelebA, each Wyner VAE consisted of an outer encoder/decoder pair and a core joint Wyner model. The outer encoder/decoder pair was introduced to pre-process raw input data. We summarized the network architectures for the Wyner models in Table 7, Table 8, and Table 10. Note that we padded zeros around the $28 \times 28$ MNIST images to make them of size $32 \times 32$.

---

[3]https://keras.io

Table 7: Network architecture for MoG experiments.

| Core Wyner VAE | |
|---|---|
| $q_\phi$'s | $p_\theta(\mathbf{x}|\mathbf{z}, \mathbf{u}), p_\theta(\mathbf{y}|\mathbf{z}, \mathbf{v})$ |
| fc-256-ReLU | fc-256-ReLU |
| fc-256-ReLU | fc-256-ReLU |
| fc-256-ReLU | fc-256-ReLU |
| (fc-10,fc-10) | fc-10 |

Table 8: Network architecture for MNIST–MNIST and MNIST–SVHN experiments.

| $\mathbf{x}, \mathbf{y}$ | | Core Wyner VAE | |
|---|---|---|---|
| Outer encoder | Outer decoder | $q_\phi$'s | $p_\theta(\mathbf{x}|\mathbf{z}, \mathbf{u}), p_\theta(\mathbf{y}|\mathbf{z}, \mathbf{v})$ |
| c5s1-32-LReLU | u3s2-128-LReLU | fc-512-LReLU | fc-512-LReLU |
| d3s2-64-LReLU | res-128 | (fc-32, fc-32)/(fc-128, fc-128) | fc-4096 |
| d3s2-128-LReLU | res-128 | | |
| res-128 | res-128 | | |
| res-128 | res-128 | | |
| res-128 | u3s2-64-LReLU | | |
| res-128 | u3s2-32-LReLU | | |
| d3s2-256 | c5s1-1-Sigmoid/c5s1-3-Sigmoid | | |

Table 9: Network architecture for MNIST quadrant prediction experiments.

| Core Wyner VAE | | |
|---|---|---|
| $q_\phi(\mathbf{z}|\mathbf{x}, \mathbf{y})$ | $q_\phi(\mathbf{u}|\mathbf{z}, \mathbf{x}), q_\phi(\mathbf{v}|\mathbf{z}, \mathbf{y})$ | $p_\theta(\mathbf{x}|\mathbf{z}, \mathbf{u}), p_\theta(\mathbf{y}|\mathbf{z}, \mathbf{v})$ |
| fc-500-ELU | fc-500-ELU | fc-500-ELU |
| fc-500-ELU | fc-500-ELU | fc-500-ELU |
| (fc-20,fc-20) | (fc-15,fc-15) | fc-392 |

Table 10: Network architecture for CelebA experiments.

| $\mathbf{x}$ (attribute) | | $\mathbf{y}$ (image) | | Core Wyner VAE | |
|---|---|---|---|---|---|
| Outer encoder | Outer decoder | Outer encoder | Outer decoder | $q_\phi$'s | $p_\theta(\mathbf{x}|\mathbf{z}, \mathbf{u}), p_\theta(\mathbf{y}|\mathbf{z}, \mathbf{v})$ |
| fc-512-LReLU | fc-512-LReLU | c5s1-32-LReLU | u3s2-128-LReLU | fc-1024-LReLU | fc-1024-LReLU |
| fc-512 | fc-40-Sigmoid | d3s2-64-LReLU | res-128 | fc-1024-LReLU | fc-1024-LReLU |
| | | d3s2-128-LReLU | res-128 | fc-1024-LReLU | fc-1024-LReLU |
| | | res-128 | res-128 | (fc-256, fc-256) | fc-512/fc-16384 |
| | | res-128 | res-128 | | |
| | | res-128 | u3s2-64-LReLU | | |
| | | res-128 | u3s2-32-LReLU | | |
| | | d3s2-256 | c5s1-3-Sigmoid | | |

**Training** We used the Adam optimizer (Kingma and Ba, 2014) with learning rate $10^{-4}$ in training MoG, MNIST–MNIST, MNIST–SVHN and CelebaA models. We used learning rate $5 \cdot 10^{-4}$ in training MNIST quadrant prediction experiment.

For MoG dataset, we trained each model for 500 epochs with batch size 100 and trained each marginal encoder for separate 50 epochs for JVAE and Wyner VAE. For MNIST–MNIST, MNIST–SVHN, and CelebA experiment, each model was trained for 100 epochs with batch size 128. For MNIST quadrant prediction experiment, each model was trained for 1000 epochs with batch size 20.

For the marginal encoder $q_\phi(\mathbf{z}|\mathbf{x})$ in JVAE or Wyner VAE, we trained the joint models by setting $\alpha_{\mathbf{x}\to\mathbf{x}} = \alpha_{\mathbf{xy}\to\mathbf{y}} = \alpha_{\mathbf{xy}\to\mathbf{x}} = \alpha_{\mathbf{y}\to\mathbf{y}} = 0$ and trained only the marginal encoder for every 50 epochs by minimizing $\overline{\mathcal{L}}_{\mathbf{x}\to\mathbf{x}}$ with freezing all the components in the joint model for 20 epochs. For MNIST–MNIST, MNIST–SVHN, and CelebA experiments, we trained $q_\phi(\mathbf{z}|\mathbf{x})$ for 1 epoch after every 1 epoch of the joint model training as suggested in Vedantam et al. (2018). Note that the outer encoder/decoder pairs in Tables 8 and 10 were trained with the core joint Wyner model, and they were fixed in the marginal encoder training. For MNIST quadrant prediction experiment, we trained the joint model and the marginal encoders $q_\phi(\mathbf{z}|\mathbf{x})$ and $q_\phi(\mathbf{z}|\mathbf{y})$ by minimizing the final objective (15) by setting all $\alpha$'s to be 1 for simplicity without fine-tuning.

We empirically observed that for some experiments the two-stage training with $\alpha_{\mathbf{xy}\to\mathbf{y}} = \alpha_{\mathbf{xy}\to\mathbf{x}} = 0$ is still effective. Note that we applied the joint training scheme and used $\alpha_{\mathbf{xy}\to\mathbf{y}}, \alpha_{\mathbf{xy}\to\mathbf{x}} > 0$ only for MNIST quadrant prediction experiment to achieve a better conditional log-likelihood performance.

