# OpenReview forum: "Wyner VAE: A Variational Autoencoder with Succinct Common Representation Learning"
_ICLR.cc/2020/Conference — Reject_

### Official Review · AnonReviewer3 · 2019-10-23
**Official Blind Review #3**

**Rating:** 6

**Review:**

This paper presents a variational auto-encoder approach for paired data variables, with a separate notion of common and individual randomness. On top of data reconstruction, Wyner's common information is added as a regularization term to promote the succinctness of the latent common representation in particular. Besides, KL divergence based constraints are proposed to encourage consistency of different modeling components. Although the consistency imposed by KL can be weak, this helps avoid making unrealistic assumptions.

To make the proposed objective function tractable, the authors make use of several standard techniques such as variational bounds and sampling-based approximations. The proposed approach can be tailored for multiple purposes, including joint/conditional generative modeling, two variable auto-encoding, style extraction and control, as illustrated in Figure 2 and Figure 3. It also supports the multi-stage training scheme, adding more flexibility. The authors did a great job in the literature review and experimental comparison. Table 1 and Table 2 demonstrate the position and distinctions of the proposed Wyner VAE approach. The comparison to the information bottleneck is also interesting.

The numerical studies are extensive. The authors demonstrate performance improvement quantitatively over several baseline competitors, although the margin is not big. With appropriate regularization, the qualitative results indeed look better subjectively.  The proposed approach stands out as a "swiss army knife" approach for two-variable VAE, which could support versatile needs. The shared and individual randomness are proved to be useful in style extraction and style control in the experiments. Overall, I feel the experiment evaluations and explanations are convincing and informative.

	• The Alice Bob example is helpful to understand the main purpose of this paper
	• The authors introduced the theoretical background of Wyner's common information --- the two contexts it arises. These texts are interesting to read, but I feel the connection to the following learning problem is weak. It would be nice to have more justifications on the choice in the context of VAE learning (optimality?), as opposed to other seemingly more natural choices. For example, the sum of mutual information I(X; Z)+I(Y; Z), or other forms of common information
	• Although the proposed approach involves many terms, in general, the presentation is good with clear notations. I am just confused about p_tilde (u,v|x,y,z) in (8). Is it simply p(u)p(v)?
	• There is a typo in (19) and (20)

**Experience Assessment:**

I have read many papers in this area.

**Review Assessment: Checking Correctness Of Derivations And Theory:**

I assessed the sensibility of the derivations and theory.

**Review Assessment: Checking Correctness Of Experiments:**

I assessed the sensibility of the experiments.

**Review Assessment: Thoroughness In Paper Reading:**

I read the paper at least twice and used my best judgement in assessing the paper.

---

> ### Author Response · Authors · 2019-11-12
> **Rejoinder**
>
> We thank the reviewer for the constructive comments. Here are our rejoinder to the comments.
>
> *** On optimality ***
> As alluded in Introduction (Section 1) and Concluding Remarks (Section 5), we believe that there exists a relation between the common information $I(X,Y;Z)$ and generalizability of the learned model (e.g., good test log-likelihood performance), and such a theoretical guarantee is currently under investigation. Although it is not the sum of mutual information $I(X;Z) + I(Y;Z)$ suggested by the reviewer, we compared the Wyner’s common information with the (variational) information bottleneck principle in Section 3 (pp.6--7) and Appendix C (p. 17), which aims to minimize a certain form of mutual information related objective.
>
> *** On $\tilde{p}(u,v|x,y,z)$ ***
> $\tilde{p}(u,v|x,y,z)$ is the conditional distribution of $u,v$ given $x,y,z$, induced by $p(z) p(u) p(v) p(x|z,u) p(y|z,v)$, meaning that $\tilde{p}(u,v|x,y,z) = p(z) p(u) p(v) p(x|z,u) p(y|z,v) / {\int p(z) p(u) p(v) p(x|z,u) p(y|z,v) du dv}$. This is why we can replace
> $$p(z) \tilde{p}(x|z) \tilde{p}(y|z) \tilde{p}(u,v|x,y,z)$$
> in eq.(8) with
> $$p(z) p(u) p(v) p(x|z,u) p(y|z,v)$$
> in eq.(9).
>
> *** On typo ***
> Thanks for pointing that out. We will update the manuscript with fixing the typo in (19) and (20).

---

### Official Review · AnonReviewer1 · 2019-10-23
**Official Blind Review #1**

**Rating:** 6

**Review:**

The paper proposed a variational autoencoder for a pair of correlated observation variables, motivated by Wyner's common information. An instantiation of the model using reverse KL divergence metric is also provided for model inference. Experimental results on simulated dataset and real image datasets show the effectiveness of the proposed generative model. The paper also provides a comprehensive appendix with technical and experimental details. Overall, the paper is technically sound and well supported by theory and experiments.

Here are a few specific comments and questions about the technical content:
1) For most of the data modeling tasks in real applications, we will most likely have one observational variable (speech segment, document or image). How would the proposed VAE model be applied when the two correlated variables are not explicitly observed or defined? It would be nice if the authors can provide some discussion on the general applicability of the  proposed model.

2) A common challenge of information theoretic VAE is the analytical intractability of divergence measure. For example in Zhao et al. 2018, the maximum mean discrepancy measure is used to approximate the divergence. It is unclear to me whether the objective function in Section 2.3 (using the reverse KL divergence measure) is analytically tractable or not. If not, what kind of approximations (e.g., sampling) did the author used?

3) In the experimental section, the authors showed that for joint and conditional distribution modeling tasks, the optimal regularization parameter might be different. This implies that for a specific task, the user is responsible to choose a proper  \lambda value. I wonder whether the authors can provide some general guidelines on how to choose this important parameter in practice, as the results seem to be highly dependent on the choice.

**Experience Assessment:**

I have read many papers in this area.

**Review Assessment: Checking Correctness Of Derivations And Theory:**

I assessed the sensibility of the derivations and theory.

**Review Assessment: Checking Correctness Of Experiments:**

I assessed the sensibility of the experiments.

**Review Assessment: Thoroughness In Paper Reading:**

I read the paper at least twice and used my best judgement in assessing the paper.

---

> ### Author Response · Authors · 2019-11-12
> **Rejoinder**
>
> We appreciate all the constructive comments. The following are the rejoinder to the technical points.
>
> *** 1) On general applicability ***
> The proposed Wyner VAE model explicitly assumes that we are given a pair of correlated observations. We have a future plan for extending our framework for (fully or partially) unpaired data. If this does not answer your question, could you elaborate on what you meant by two correlated variables are not explicitly observed?
>
> *** 2) On tractability of KL divergence measure***
> The (reverse) KL divergence is not analytically tractable, so we used Monte Carlo sampling for approximation as you mentioned, which is the standard technique in the VAE literature.
>
> *** 3) On a general guideline for choosing $\lambda$ ***
> As typically done in any statistical method, $\lambda$ can be chosen by cross-validation. For example, given a split of dataset into three parts {training, test, validation}, we train a model with a set of $\lambda$ choices using training dataset, and find the best achieved performance for each $\lambda$ based on the validation dataset. The final $\lambda$ can then be chosen based on the performance on the test split.

---

### Official Review · AnonReviewer2 · 2019-10-30
**Official Blind Review #2**

**Rating:** 3

**Review:**

This paper presents a method for learning latent-variable models of paired data that decomposes the latent representation into common and local representations. The approach is motivated by Wyner’s common information, and is made tractable through a variational formulation. On experiments with MoG, MNIST, SVHN, and CelebA, the paper shows that penalizing the complexity of the common representation can improve style-content disentanglement and conditional sampling.

While I found the formulation and motivation from Wyner’s common information and Cuff’s channel synthesis interesting, the resulting model and experiments were unconvincing. There are many terms in the Wyner model proposed, and there are no ablations demonstrating which terms are important and which are not (e.g. do you need both L_xx and L_xy?). On the toy MoG experiment, the common information is known but trained models do not recover the right amount of information. For representation learning, accuracy decreases as the penalty on the common information increases, and for NLL, the joint and conditional NLL is often similar to existing work (CVAE and JVAE). The main win appears to be for style-content disentanglement, but the results there are qualitative and often change the content when only style is changed. It’s also puzzling as to why CVAE overfits so severely in a subset of the experiments. Without a more thorough evaluation of what terms in the loss matter, and showing that the technique recovers something like Wyner’s common information (by extracting the right information on a toy model), I cannot recommend this paper for acceptance.

Minor comments:
* Eqn 1: the constraint “Subject to X-Z-Y” is confusing. Do you mean X-Z-Y is a Markov network (undirected) and not a Markov chain (directed), in which case X -> Z <- Y does not correspond to X-Z-Y? (This is addressed in Eqn 3-4, but should be fixed here)
* How is Wyner’s common information related to the multivaraite mutual information I(X; Z; Y)?
* When describing distributed simulation, please include the variables and objective like you do for common information
* s/Markovity condition/independence assumption?
* Why are both losses in Eqn 4 and Eqn 5 needed? Couldn’t you use either to train q(z|x)?
* Eqn 10 (and most of your objectives) are still intractable due to the q(x)/q(x,y) terms, you should note that it is constant and dropped from the objective
* “Style control”: could you define what you mean by style in this context? Prior work could likely also do “style control” by e.g. interpolating subsets of dimensions.
* Related work: https://openreview.net/forum?id=rkVOXhAqY7 (CEB) that may result in a similar objective as Wyner’s common info
* When comparing to JVAE/JMVAE, it seems like the main difference is suing a latent-variable in the decoder, but the framework is still the same. It’d be useful to spend more time comparing/contrasting with this prior work.
* Fig 4: would be useful to have a picture of the samples (right now they’re just in appendix)
* Fig 4: in the data generating process, I believe I((X, Y); Z) = ln(5) = 1.6 nats, but none of your models converge to rates around there. Why not?
* Unlike other approaches (JVAE, CVAE) you have additional terms in your loss for conditional prediction at training time. It seems like these may be giving you gains, and it’d be useful to perform ablations over the terms in Eqn 15 to figure out what is impacting performance in Fig 4. E.g. if you set \alpha_{x -> y} and \alpha{y -> x} to 0
* Why do the CVAE models overfit? Have you tried optimizing the variational parameters on the test set if it’s due to amortization gap?
* Fig 5b: what’s the variance you’re plotting? In representation learning, we often just care about downstream accuracy, and it looks like \lam = 0  performs best there.
* Table 3: isn’t this showing that \lam = 0 works the best for joint NLL and close to best for conditional NLL on the MoG task? What’s the discrepancy with Fig 5 where conditional NLL is highly dependent on \lambda for MNIST?
* Fig 6 (a1-f1): for this MNIST add 1 dataset, the only common info should be the label. But it looks like the labels do change for non-zero values of lambda (b1 there’s a 2 -> 7), c1 1 -> 9)
* Table 4: would be useful to train CVAE yourself, as the small differences in numbers could just be due to tuning / experimental setup. You also should bold everything that’s within stderr (i.e. \lambda = 0 and \lambda = 0.15 are equally good)
* Fig 9: would be useful to include CVAE, and Wyner VAE in this plot, i.e. is it only using the mode information in the latent variable when doing conditional sampling?
* CelebA results look interesting, but there’s been other work on generation from attributes that presents more visually compelling results, e.g. https://arxiv.org/abs/1706.00409

**Experience Assessment:**

I have published one or two papers in this area.

**Review Assessment: Checking Correctness Of Derivations And Theory:**

I assessed the sensibility of the derivations and theory.

**Review Assessment: Checking Correctness Of Experiments:**

I assessed the sensibility of the experiments.

**Review Assessment: Thoroughness In Paper Reading:**

I read the paper at least twice and used my best judgement in assessing the paper.

---

> ### Author Response · Authors · 2019-11-10
> **One question seems to be broken.**
>
> We thank for your critical reviews and constructive comments. While we are working on the answers, we found that part of one of your question was somehow deleted. ("* s/Markovity condition/independence assumption?") Could you restate your question?

---

> > ### Comment · AnonReviewer2 · 2019-11-15
> > **clarification**
> >
> > I was just using the sed notation (s/original string/replacement string) to denote a suggested text replacement of changing "Markovity condition" which is a term I've never heard of to "independence assumption" which is more common.

---

> > > ### Author Response · Authors · 2019-11-15
> > > **Thanks for your clarification.**
> > >
> > > We updated the manuscript by updating the term "Markovity condition" to "Markov assumption", which seems to be the standard terminology. (See https://en.wikipedia.org/wiki/Markov_property.)

---

> ### Author Response · Authors · 2019-11-12
> **First rejoinder (1/3)**
>
> The major concerns raised are 1) the lack of an ablation study on the objective function and 2) that the Wyner’s common information was not recovered in the toy MoG dataset. We are currently running some additional experiments to address these, and the results and the answers to the related comments will be posted soon.
>
> To make discussion more interactive, here is our first rejoinder for the rest of the comments.
>
> ---------------------------------------------------------------------------------------------------------------------------
> * Eqn 1: the constraint “Subject to X-Z-Y” is confusing. Do you mean X-Z-Y is a Markov network (undirected) and not a Markov chain (directed), in which case X -> Z <- Y does not correspond to X-Z-Y? (This is addressed in Eqn 3-4, but should be fixed here)
>
> >>> We will update the manuscript with the standard notation “$X\to Z\to Y$” for a Markov chain (which is equivalent to “$Y\to Z\to X$”) to avoid confusion.
>
> ---------------------------------------------------------------------------------------------------------------------------
> * How is Wyner’s common information related to the multivariate mutual information I(X; Z; Y)?
>
> >>> There is no connection; refer to (El Gamal and Kim, 2011) Network Information Theory for more details on Wyner’s common information.
>
> ---------------------------------------------------------------------------------------------------------------------------
> * When describing distributed simulation, please include the variables and objective like you do for common information
>
> >>> The fundamental limit of the distributed simulation is characterized by the identical optimization problem eq. (1) as in the channel synthesis case. We will articulate this point clearly in the revised manuscript.
>
> ---------------------------------------------------------------------------------------------------------------------------
> * s/Markovity condition/independence assumption?
>
> >>> Part of the question was somehow deleted. Could you restate your question?
>
> ---------------------------------------------------------------------------------------------------------------------------
> * Why are both losses in Eqn 4 and Eqn 5 needed? Couldn’t you use either to train q(z|x)?
>
> >>> We can train q(z|x) only with one of (eq. 4) or (eq. 5). As you suggested, we will update an additional ablation study on the effect of having one of (eq. 4, 5) or both.
>
> ---------------------------------------------------------------------------------------------------------------------------
> * Eqn 10 (and most of your objectives) are still intractable due to the q(x)/q(x,y) terms, you should note that it is constant and dropped from the objective
>
> >>> We will clarify in the manuscript that the objective can be approximated by Monte Carlo sampling and up to entropy of data distributions $q(x)$ and/or $q(x,y)$, which are constant.
>
> ---------------------------------------------------------------------------------------------------------------------------
> * “Style control”: could you define what you mean by style in this context? Prior work could likely also do “style control” by e.g. interpolating subsets of dimensions.
>
> >>> In our context, “style” information of $X$ and $Y$ are defined as the local latent variables $U$ and $V$, respectively, where $(Z,U,V)$ is drawn from the encoders $q(z|x,y)q(u|z,x)q(v|z,y)$. On the decoder side, $X$ is a function of $(Z,U)$ and $Y$ is a function of $(Z,V)$, which implies that $U$ and $V$ can be interpreted as the remaining information left behind after excluding the “common part” $Z$. We remark that the Wyner model aims to find the succinct common part $Z$ for better generalization and better generation performance.
>
> ---------------------------------------------------------------------------------------------------------------------------
> * Related work: https://openreview.net/forum?id=rkVOXhAqY7 (CEB) that may result in a similar objective as Wyner’s common info
>
> >>> To our understanding, CEB is a special case of information bottleneck (IB) principle, in which the latent variable sits outside as “$Z\to X\to Y$”. Please refer to Section 3 on how IB principle differs from the proposed Wyner VAE (“$X\to Z\to Y$”).

---

> ### Author Response · Authors · 2019-11-12
> **First rejoinder (2/3)**
>
> * When comparing to JVAE/JMVAE, it seems like the main difference is suing a latent-variable in the decoder, but the framework is still the same. It’d be useful to spend more time comparing/contrasting with this prior work.
>
> >>> The main difference of JVAE/JMVAE from Wyner VAE is that the shared latent variable $Z$ [which we denote $W$ in the manuscript to avoid confusion with our $Z$] can deterministically generate $X$ without explicit local latent variable $U$ and $V$. Hence, in JVAE/JMVAE, $X$ and $Y$ are each functions of $Z$, while in Wyner VAE, $X$ is a function of $(U,Z)$ and $Y$ is a function of $(V,Z)$. The role of $Z$ in JVAE/JMVAE is identical to that of $Z$ in the VAE except that $(X,Y)$ are jointly generated. But in Wyner VAE, the role of $Z$ is the succinct common representation of $X$ and $Y$, not the entirety of the randomness that generates both of them. We will emphasize this contrast in the revised manuscript.
>
> ---------------------------------------------------------------------------------------------------------------------------
> * Fig 4: would be useful to have a picture of the samples (right now they’re just in appendix)
>
> >>> We will update the manuscript with the picture of the samples (Fig 9. in Appendix).
>
> ---------------------------------------------------------------------------------------------------------------------------
> * Fig 4: in the data generating process, I believe I((X, Y); Z) = ln(5) = 1.6 nats, but none of your models converge to rates around there. Why not?
>
> >>> As you pointed out, $I(X,Y;Z)=\log 5$ in this toy example, as $Z$ can be exactly recovered from $(X,Y)$. We will update this answer with the additional experimental results.
>
> ---------------------------------------------------------------------------------------------------------------------------
> * Unlike other approaches (JVAE, CVAE) you have additional terms in your loss for conditional prediction at training time. It seems like these may be giving you gains, and it’d be useful to perform ablations over the terms in Eqn 15 to figure out what is impacting performance in Fig 4. E.g. if you set \alpha_{x -> y} and \alpha{y -> x} to 0
>
> >>> We will post results from an ablation study soon.
>
> ---------------------------------------------------------------------------------------------------------------------------
> * Why do the CVAE models overfit? Have you tried optimizing the variational parameters on the test set if it’s due to amortization gap?
>
> >>> On why CVAE model overfits:
> If the conditioning variable is discrete, CVAE learns a separate VAE for each conditional variable value. If the conditioning variable is over a large alphabet or even in a high-dimensional space as in our setting, however, CVAE requires a huge number of samples. Using a neural network for the conditioning variable may alleviate the large alphabet problem, but it still ignores the distribution of the conditioning variable, which may give an additional regularization and thus may lead to better generalization (Shu et al. 2017).
>
> >>> On fine-tuning for the variational parameters:
> We used the standard Gaussian parameterization for encoder/prior/decoder as in the original CVAE paper, and did not optimize the variational parameters on the test across all the models as a fair comparison.
>
> ---------------------------------------------------------------------------------------------------------------------------
> * Fig 5b: what’s the variance you’re plotting? In representation learning, we often just care about downstream accuracy, and it looks like \lam = 0  performs best there.
>
> >>> The variance in Fig.5b is the per-pixel variance in the generated images as mentioned in the main text. This figure is to show the trade-off between the “correctness” of the conditional path (in terms of the label accuracy) and the variety of generated images (in terms of the per-pixel variance), as we vary the regularization parameter $\lambda$. In our context, we are interested in joint and conditional generation, not classifying the label of data points based on the inferred latent variables. In Fig. 6 and Fig. 12, JVAE generated samples with almost no variations, while Wyner VAE generated samples of various styles.

---

> ### Author Response · Authors · 2019-11-12
> **First rejoinder (3/3)**
>
> * Table 3: isn’t this showing that \lam = 0 works the best for joint NLL and close to best for conditional NLL on the MoG task? What’s the discrepancy with Fig 5 where conditional NLL is highly dependent on \lambda for MNIST?
>
> >>> As you pointed out, the gain from $\lambda=0.15$ is marginal compared to $\lambda=0$ case. We claim that the regularization may have more impact on the performance if the underlying data has inherently larger local information compared to the common information.
> To get an intuition, let’s consider two extreme cases. First, suppose that the data variables $(X,Y)$ are simply a function of another random variable $Z$. In this case, the local variables in Wyner VAE are unnecessary and regularization on the common information part $I(X,Y;Z)$ may only hinder learning, as it may push the common information to the local parts unncessarily. On the other extreme, suppose now that $(X,Y)$ are not correlated in the sense that there exist independent random variables $U$ and $V$ such that $X$ and $Y$ are functions of $U$ and $V$, respectively. In this case, the common variable $Z$ in Wyner VAE must not capture any information for $(X,Y)$, and thus the regularization with a larger $\lambda>0$ will always make learning better.
> In practice, the situation is somewhere in between the two extremes. We suspect that the MoG dataset and the MNIST quadrant prediction dataset are closer to the first extreme, while the MNIST--MNIST and MNIST--SVHN dataset are closer to the second extreme. To support and illustrate this argument, we are currently running experiments with MoG dataset with larger variances on the local variables in the data generating process.
>
> ---------------------------------------------------------------------------------------------------------------------------
> * Fig 6 (a1-f1): for this MNIST add 1 dataset, the only common info should be the label. But it looks like the labels do change for non-zero values of lambda (b1 there’s a 2 -> 7), c1 1 -> 9)
>
> >>> Note that while the images in Fig. 6 (b1, c1) may be read as 7 and 9, they also look close to the supposedly correct labels 2 and 1, respectively. In our case, the continuous latent space is partitioned into several classes, such that similar images are to be around each other as in the standard VAE, and the label mismatch is inevitable due to this continuous parameterization. Note that no label information is utilized in the set of experiments.
>
> ---------------------------------------------------------------------------------------------------------------------------
> * Table 4: would be useful to train CVAE yourself, as the small differences in numbers could just be due to tuning / experimental setup. You also should bold everything that’s within stderr (i.e. \lambda = 0 and \lambda = 0.15 are equally good)
>
> >>> We trained CVAE ourselves with two network designs; one is from the original CVAE paper (Sohn et al. 2016), a 2-layer MLP with hidden units [1000,1000] and relu activations followed by 200 Gaussian latent variables, while the other is from our paper which uses a 2-layer MLP with hidden units [500, 500] and elu activations followed by 50 Gaussian latent variables. We observed that overfitting happened rather quickly after several epochs as we reported, and the lowest achieved log likelihoods we produced were not better than the number from (Sohn et al. 2016).
> We will update the manuscript with proper bolds as you pointed out.
>
> ---------------------------------------------------------------------------------------------------------------------------
> * Fig 9: would be useful to include CVAE, and Wyner VAE in this plot, i.e. is it only using the mode information in the latent variable when doing conditional sampling?
>
> >>> Regarding Fig 9, we did not list the 2d-projection plots from CVAE and Wyner VAE, as they look almost the same as the training data (Fig.9(a)). We will add them in the main text.
> In the conditional sampling, we take a random sample from the encoder using both mean and variance.
>
> ---------------------------------------------------------------------------------------------------------------------------
> * CelebA results look interesting, but there’s been other work on generation from attributes that presents more visually compelling results, e.g. https://arxiv.org/abs/1706.00409
>
> >>> Thanks for the reference. We plan to work on improving more visually compelling results in the future.

---

> ### Author Response · Authors · 2019-11-15
> **On ablation study**
>
> Due to the limited time frame, we were not able to run an enough number of experiments for ablation study.
> Although we have a partial result from the MNIST prediction task, we have not updated the paper with this result yet, as we do not want to make a possibly false conclusion from a single dataset. We will perform the ablation study on the objectives across other datasets to include the results in a future revision.
> We conjecture that the effect of the obectives may be different depending on the nature of datasets, e.g., for example, the complexity of common information structure, and the amount of local randomness.
>
> Here is the partial result from the MNIST quadrant prediction task.
>
>
> 		|	J+M+C ($\lambda=0$)	|	J+M+C ($\lambda=0.15$)	|	J+M, ($\lambda=0.15$)	|	J+C ($\lambda=0.15$)
> ------------------------------------------------------------------------------------------------------------
> mean	|	87.74	43.87		|	89.66	43.85			|	89.57	44.76		 	|	91.15	44.35
> std.p	|	0.05	0.03		                |	0.03	0.03			                |	0.06	0.06			                |	0.08	0.06
>
> - J, M, C are the abbreviations of joint, marginal, and conditional, respectively.
> - All the weights were uniformly 1 or 0.
> - The two numbers correspond to joint and conditional negative log likelihood, respectively.
> - All the experiments were performed eight times and the results were averaged over different seeds.
> - This result shows that J+M+C performs almost equally best for both $\lambda=0$ and $\lambda=0.15$.
> - However, we may need to explore different choices of $\lambda$ to correctly assess the effect.

---

> ### Author Response · Authors · 2019-11-15
> **Few remaining points**
>
> * On additional experiments on explaining the discrepancy between MNIST--MNIST add-1 and synthetic dataset
> First, we think the synthetic dataset shows some improvement with $\lambda>0$, yet the amount is marginal. However, the effect does not clearly appear in the MNIST quadrant prediction task, and the reasoning was addressed in the final comment above. Please refer to that.
> Also, we were not able to find a good synthetic dataset that can capture such an effect by changing the variance in the local variables in the data generating process.
>
>
> * On estimating mutual information
> We attribute the mismatch to the Gaussian parameterization of the underlying discrete latent variable and variational approximation in the model. Please note that our main claim is not to estimate the exact common information, but to improve generative performance by introducing the local variables and controlling $\lambda$, which were shown in the set of experiments.
>
>
> * On Fig. 9 with additional samples
> We updated Fig. 9 with additional samples from Wyner VAE, VCCA-private, and CVAE.

---

### Official Review · AnonReviewer4 · 2019-11-04
**Official Blind Review #4**

**Rating:** 6

**Review:**

The paper proposes Wyner VAE, a variational autoencoder for a five-variable graphical model. The five variables consist of two observable views, X and Y, with a shared latent variable Z, while the two other latent variables U and V control the randomness of X and Y, respectively, independent of others. The overall objective is motivated by Wyner's mutual information minimization. Using a few common techniques to bound KL divergence, the paper arrives at a few terms that are reminiscent to the variational lower bounds, including terms that constraints the hidden variable to be similar to the prior, and two cross-modal reconstruction terms. The paper includes a comparison between similar models, both in terms of formulation and experiments. The results show that Wyner VAE has certain nice properties that is lacking in other models.

I am giving a score of 6. The formulation and the lower bound for learning are the strengths of the paper. The lack of discussion on some of the common problems for VAEs and the experiments are the weakness of the paper.

The paper deserves credit to motivate the problem from Wyner's viewpoint. However, after the derivation, the framework falls back to common VAEs, and is not very different from VCCA-private. Comparing Wyner VAE and VCCA-private, I can see why it is hard to justify the difference of the two based on the formulations, but I also find it unconvincing from the experiments to conclude that one is better than the other, especially when there are many hyperparameters to tune. The paper should state this clearly when comparing the two.

In terms of the common problems for VAEs, the paper does not address a few degenerate cases. The first case is where all the information gets pushed to the shared latent variable Z, leaving little private information for U and V. The second case is that little information about both views gets pushed to the shared latent variable Z, while the decoders are doing all the heavy lifting to reconstruct the observables. To avoid these degenerate cases, more constraints might be needed for the models, such as the ones proposed in (Hsu et al., 2017). Finally, as with most VAE models, the paper does not discuss how well lower bounds approximates the likelihood and the resulting learned representation when the gap between the likelihood and the objective is large.

The experiments in the paper are somewhat lacking. Figure 7 is where the capability of the model is fully demonstrated: a clear distinction between the shared and the private views are shown. It would be great if the paper can include experiments on a real-world multi-view data set, such as images paired with texts, speech paired with images, speech paired with video, etc.

Unsupervised learning of disentangled and interpretable representations from sequential data
Wei-Ning Hsu, Yu Zhang, James Glass
Neurips, 2017


**Experience Assessment:**

I have read many papers in this area.

**Review Assessment: Checking Correctness Of Derivations And Theory:**

I carefully checked the derivations and theory.

**Review Assessment: Checking Correctness Of Experiments:**

I carefully checked the experiments.

**Review Assessment: Thoroughness In Paper Reading:**

I read the paper thoroughly.

---

> ### Author Response · Authors · 2019-11-12
> **Rejoinder**
>
> We appreciate all the constructive comments. The following are the rejoinder to the technical points.
>
> *** On Wyner VAE vs. VCCA-private ***
> As you pointed out, Wyner VAE and VCCA-private share the same generative model (decoder model), but the difference lies in the encoder model; see Appendix C. Wyner VAE assumes an encoder model $q(z,u,v|x,y)=q(z|x,y)q(u|z,x)q(v|z,y)$, which captures a consistent conditional independence to the Markovity condition in the decoder model. On the other hand, VCCA-private assumes an encoder model $q(z,u,v|x,y)=q(z|x)q(u|x)q(v|y)$, which is a rather ad-hoc conditional independence structure. We believe that it inherently implies a limited generative performance, which is demonstrated with MNIST--MNIST add-1 dataset; see Fig. 6 (f1, f2).
>
> *** On degenerate cases in Wyner VAE ***
> The two degenerate cases you pointed out are addressed in the current Wyner VAE model as follows. The regularization weight $\lambda>0$ on the common information $I(X,Y;Z)$ is to avoid the first degenerate case where “all the information gets pushed to the shared latent variable $Z$, leaving little private information for $U$ and $V$”. We also remark that JVAE/JMVAE models this degenerate case, lacking the local (private) random variables $U$ and $V$—the shared latent variable $Z$ captures all information to generate both $X$ and $Y$, and it leads to a poor generalization performance; see Table 3. The second degenerate case may occur when we put too large regularization weight $\lambda$ and/or when the amount common information $I(X,Y;Z)$ is relatively small compared to the local information. For the latter case, we note that the conditional losses in eq. (14) can help the model avoid such degeneracy, as the shared variable $Z$ is forced to capture the common information of $X$ and $Y$ to make the conditional losses small. We will add an additional experiments demonstrating this effect.
>
> *** On the lack of discussion regarding variational approximation ***
> Thanks for your comment. We will address the problem on the variational approximation and the resulting learned representation in the future work.
>
> *** On experiment with real-world dataset ***
> Thanks for the pointer. In the current manuscript, we focused on showing the validity and the potential of our framework. We plan to extend the fundamental concepts introduced in the current work to practical applications such as frame interpolation/ extrapolation for single- or multi-view videos per your suggestion.

---

> > ### Comment · AnonReviewer4 · 2019-11-14
> > **Some feedback**
> >
> > Thanks for the response.
> >
> > VCCA-private assumes an encoder model , which is a rather ad-hoc conditional independence structure.
> > --> I am not trying to defend VCCA, but "ad-hoc" is a rather subjective property.
> >
> > We believe that it inherently implies a limited generative performance, which is demonstrated with MNIST--MNIST add-1 dataset; see Fig. 6 (f1, f2).
> > --> The experiments can only confirm, but cannot really prove/disprove any the claims. The decoders can decide to ignore some of the variables as well. I guess it's better to say that the decoders in Wyner VAE is strictly more general than the ones in VCCA?
> >
> > On degenerate cases in Wyner VAE ... We will add an additional experiments demonstrating this effect.
> > --> It might be good to talk the the two extreme cases (lambda=0 and lambda -> inf) in the paper. Maybe it's better to construct synthetic distributions where we can actually measure the mutual information and see the degenerate cases happening while tuning lambda. These degenerate cases are long-standing problems for most multiview VAEs, and I don't think Wyner VAE actually solves this problem. Asking for a solution in this paper is probably too much, but the authors should point out the weaknesses in the paper.
> >
> > I have also read other reviews and comments. I have mixed feelings about CelebA results, especially the term "Not cherry-picked." The CelebA results really do not show any additional information, and it is hard to say which is more visually compelling (unless subjective tests are done). It suffices to say that Wyner VAEs work as designed.

---

> > > ### Author Response · Authors · 2019-11-15
> > > **Replies**
> > >
> > > We appreciate your feedback.
> > >
> > > - In the revised manuscript, we avoided using the terminology "ad-hoc" as you pointed out.
> > > We elaborated that our encoder structure is naturally derived by the decoder structure $p(z)p(u)p(v)p(x|z,u)p(y|z,v)$, compared to the VCCA-private's encoder structure.
> > >
> > > - While decoders can ignore certain variables, the semantic meaning of each $Z, U, V$ may become different by assuming different encoder architecture. For example, the local variable $U$ in Wyner VAE captures the remaining information of $X$ excluding the "common" part $Z$ with the help of the explicit probabilistic encoder $q(u|z,x)$.
> > > However, VCCA-private encodes $x$ by two conditionally independent random variables $U$ and $Z$ by using the encoders $q(u|x)$ and $q(z|x)$, and this does not apparently capture the common--local information decomposition in Wyner VAE. However, we agree that the experimental results can only confirm this intuition, not proving/disproving one's argument.
> > >
> > > - We appreciate your comment regarding the degenerate cases. In the revised manuscript, we added a paragraph for the degenerate cases that can happen in Wyner VAE.
> > > We summarize the argument here briefly.
> > > (1) The first degenerate case where $Z$ captures the entire information is captured by the joint VAE (JVAE) model. We showed the advantages of Wyner VAE (even with $\lambda=0$) compared to the degenerate case (i.e., the joint VAE), showing that even having the local latent variables seems to help circumvent the degeneracy.
> > > (2) The second degenerate case is where $Z$ does not capture any information, where it may happen as $\lambda\to\infty$. This degeneracy is demonstrated in, for example, Fig. 5 (right); as $\lambda$ increases, the desired label information gets lost.
> > >
> > > - Due to the limited time frame, we were not able to come up with a toy example where we can recover the exact common information. We will add some examples in the future revision.
> > >
> > > - As you pointed out, the CelebA results are auxiliary, and it is just to confirm that Wyner VAE has some advantages even in this degenerate dataset, i.e., Wyner VAE can find a good representation for the attribute information (Fig. 17) and also can extract style information (Fig. 19, 20). We will remove the misleading term "cherry-picked" and tone down the paragraph.

---

### Comment · Area_Chair1 · 2019-11-14
**Reviewers, and comments on the author responses?**

Dear Reviewers, thanks for your thoughtful input on this submission!  The authors have now responded to your comments.  Please be sure to go through their replies and revisions.  If you have additional feedback or questions, it would be great to get them this week while the authors still have the opportunity to respond/revise further.  Thanks!

---

### Author Response · Authors · 2019-11-15
**Final comments**

We appreciate all the reviewers for their thoughtful and constructive comments. We improved and uploaded the manuscript based on your input. Here is our final response to the main questions.

* Main goal and contributions

The goal of this paper is to propose a new probabilistic model based on two information theoretic problems (distributed simulation and channel synthesis), to achieve a good performance in joint and conditional generation tasks.
We apply the idea of learning succinct common representation to design a new generative model for a pair of correlated variables: seeking a succinct representation $Z$ in learning the underlying distribution based on its sample may also help reduce the burden on the decoder's side and thereby achieve a better generative performance.

The performance gain of Wyner VAE compared to other models can be attributed to the following two factors:

1) Introduction of the local latent variables $U$ and $V$.
- Removing the connections from/to $U$ and $V$, Wyner VAE boils down to JVAE/JMVAE, where $Z$ is supposed to encode all the information of both $X$ and $Y$.
Across the experiments, we showed that Wyner VAE with $\lambda=0$ clearly outperforms the degenerate case, JVAE.
- There also exists an existing model VCCA-private, but Wyner VAE assume a more natural encoder structure conforming to the decoder structure it assumes.

2) Introduction of $\lambda$ control to promote to find a more succinct representation, which may lead to better generalization.
- We note that the effect of $\lambda$ may depend on the nature of the dataset. For MoG, MNIST--MNIST add-1, and MNIST--SVHN experiments, Wyner VAE improved at a positive $\lambda$, on top of introducing $U$ and $V$.
However, in the MNIST quadrant prediction task, $\lambda$ does not show a clear improvement, while Wyner VAE with $\lambda=0$ achieves a clear improvement over JVAE.
We conjecture that this may be attributed to the nature of dataset; the common information structure of the MNIST quadrant prediction dataset may be much more complex than the previous examples, and $\lambda>0$ may not show clear advantages compared to $\lambda=0$. We emphasize again, however, that Wyner VAE with $\lambda=0$ has an ability to capture a more succinct common representation than the degenerate model JVAE.

We articulated this two-stage gain explicitly in the revised manuscript.


* On estimating common information

We would like to remark that the main goal of this paper is not to exactly estimate the Wyner's common information.
Instead, our main claim is that the Wyner VAE built upon the information theoretic intuition may lead to better generalization performance.
We agree, however, that it would be interesting to see a toy example where the Wyner's common information is analytically obtained and Wyner VAE can truly recover the quantity. We were not able to add such an example due to the limited time frame, but will investigate the direction for the future revision.


* On degenerate cases

Two degenerate cases may arise in Wyner VAE.
The first case is where the common variable $Z$ captures all information of $(X,Y)$, while $U$ and $V$ capture none.
Joint VAE (JVAE)~\citep{Vedantam--Fischer--Huang--Murphy2018TELBO} and joint multimodal VAE (JMVAE)~\citep{Suzuki--Nakayama--Matsuo2016JMVAE} inherently assume this degenerate case, as it is discssued in the next section.
Wyner VAE is able to avoid such degeneracy by explicitly having the local variables and controlling the common regularization parameter $\lambda>0$.
On the other extreme, $Z$ may capture no information, while $U$ and $V$ capture all the information of $X$ and $Y$, respectively. It may happen in Wyner VAE if the regularization parameter $\lambda$ is too large.
To avoid the degeneracy, we need to choose a proper $\lambda$ by cross-validation.


* On ablation study
Please refer to our separate response to Reviewer 2's comments.

---

### Decision · Program_Chairs · 2019-12-19

**Decision:**

Reject

**Comment:**

This paper adds a new model to the literature on representation learning from correlated variables with some common and some "private" dimensions, and takes a variational approach based on Wyner's common information.  The literature in this area includes models where both of the correlated variables are assumed to be available as input at all times, as well as models where only one of the two may be available; the proposed approach falls into the first category.  Pros:  The reviewers generally agree, as do I, that the motivation is very interesting and the resulting model is reasonable and produces solid results.  Cons:  The model is somewhat complex and the paper is lacking a careful ablation study on the components.  In addition, the results are not a clear "win" for the proposed model.  The authors have started to do an ablation study, and I think eventually an interesting story is likely to come out of that.  But at the moment the paper feels a bit too preliminary/inconclusive for publication.